# Revitalizing Intangible cultural heritage via derivative design: A focus on chinese woodblock printing

**Shao-Feng Wang**[1,2]*, **Chun-Ching Chen**[3]

1 Doctoral Program in Design, College of Design, National Taipei University of Technology, Taipei, Taiwan,
2 Department of Product Design, Xiamen Academy of Arts and Design, FuZhou University, Xiamen, China,
3 Department of Interaction Design, College of Design, National Taipei University of Technology, Taipei, Taiwan

* t09302@fzu.edu.cn

## Abstract

Traditional safeguarding of intangible cultural heritage often prioritizes the preservation of craftsmanship, while allocating less attention to the innovation and activation of derivatives. This investigation extracts the identifiable intellectual property from Chinese woodblock New Year pictures, establishes a design framework to transition from two-dimensional to three-dimensional elements, and integrates generative design to examine the viability and adaptability of generative Design methods. Specifically, employing Chinese woodcut New Year pictures as a pivotal case, design constituents are elicited through interval questionnaires and on-site surveys. Subsequently, employing a comprehensive experiential approach and user journey mapping, a compendium of generative Design methods is distilled. Augmenting this, recommendations are formulated by considering generative design as a variable. Conclusively, through a dependent sample T-test, it is discerned that generative design effectively enhances both the quantity and caliber of design propositions. This inquiry dissects the plausibility of the generative Design approach for Chinese woodblock New Year pictures, extrapolating a blueprint for said derivatives and unearthing three-dimensional design concepts within the methodological framework. Ultimately, this endeavor facilitates the dynamic preservation and widespread propagation of intangible cultural heritage, concurrently bridging the chasm between planar and three-dimensional realms through innovative paradigms.

## 1. Introduction

Intangible Cultural Heritage (ICH) plays a crucial role in cultural diversity, social cohesion, and economic development. On one hand, protecting Intangible Cultural Heritage helps enhance cultural inheritance, national confidence, and tourism economy; on the other hand, it can also promote sustainable social and economic development [1]. However, the process of protecting and inheriting Intangible Cultural Heritage faces the following challenges: (1) conservative themes that fail to meet the needs of the new era; (2) the high difficulty of traditional craftsmanship hinders modernization; (3) severe industry homogenization affects cultural

**Data availability statement:** All data generated or analyzed in the course of this study are included in this published article and its supplementary file.

**Funding:** The author(s) received no specific funding for this work.

**Competing interests:** The authors have declared that no competing interests exist.

dissemination; (4) the scarcity of derivatives impedes commercial communication. Therefore, it is evident that innovative protection models and traditional approaches are necessary to endow traditional culture with new vitality and value [2].

The practical application of Intangible Cultural Heritage in design innovation mostly remains at the stage of aesthetic form, lacking systematic innovation from the perspective of cultural elements and craftsmanship [3]. Current shortcomings in design innovation for Intangible Cultural Heritage protection are mainly reflected in the following aspects: (1) innovation only in form, without a profound understanding of the importance of cultural values [4]; (2) excessive pursuit of marketization and commercialization, leading to homogenization of existing products. While commercialization can expand the scope of dissemination, the single-minded pursuit of economic benefits weakens the original cultural and historical significance of Intangible Cultural Heritage [5]; (3) limitations of design innovation methods. The failure to adopt modern design thinking and tools leads to creative exhaustion; (4) the limitations of dissemination media affect the range and acceptance of Intangible Cultural Heritage [6]. In conclusion, to address the limitations of Intangible Cultural Heritage protection methods and dissemination channels, researchers should explore innovative thinking and methods from the root and apply them to specific cases for refinement and iteration. In the field of design, updating and applying design methods is the source of innovation.

This study uses Zhangzhou Woodblock New Year Pictures as an example to explore methods of designing derivative products and innovative protection models for intangible cultural heritage.In May 2006, Chinese woodblock New Year pictures were enshrined as part of the inaugural selection of national intangible cultural heritage representative endeavors. The custodian of the Chinese woodblock New Year pictures legacy at the provincial level is Yan Chaojun, the seventh-generation torchbearer of the Yan's woodblock New Year pictures heritage. Conjoined with the woodblock New Year pictures of "Taohuawu in Jiangsu," "Mianyang in Sichuan," and "Yangliuqing in Tianjin," Chinese woodblock New Year pictures collectively constitute the quartet of eminent Chinese woodblock New Year picture production locales. This artistry predominantly radiates from the precincts of southern Fujian and the Lingnan region. By means of cultural dissemination, it has expansively carried the artistic essence of Chinese woodblock New Year pictures to enclaves such as Hong Kong, Macao, Taiwan, and Southeast Asia. As of the present juncture, the principal quandaries confronting Chinese woodblock New Year pictures are:

(1) Traditional intangible cultural heritage still retains its vitality, conveying national culture and historical symbols. Its craftsmanship showcases how technology can drive cultural dissemination, while its cultural traits reflect regional customs, fostering intercultural exchange and historical inheritance. However, with the advancement of technology and the impact of modern culture, traditional woodblock prints face challenges such as limited mediums of dissemination, gaps in transmission paths, and insufficient commercial development. Therefore, it is urgently necessary to utilize derivative design to provide more diverse dissemination platforms, wider channels of communication, and approaches that better align with contemporary societal needs.

(2) There are few employees in Woodblock New Year pictures, and they need help with skill inheritance and cul-tural protection.

(3) The complex process of woodblock New Year pictures leads to high costs, which makes it difficult to meet the commercial needs of batches.

(4) Industry homogenization is serious, and the regional characteristics must be excavated and innovated urgently.

In 2004, China became a signatory to UNESCO's Convention for the Safeguarding of Intangible Cultural Heritage, thereby embarking on a progressive journey toward safeguarding intangible cultural heritage under the stewardship of governmental initiatives. Subsequently, an inventory encompassing national, provincial, and local non-legacy representatives was established, concomitantly introducing the concepts of comprehensive, rescue-based, living, and productive protection realms. These endeavors have yielded substantial achievements [7].

Historically, the exploration of safeguarding and carrying forward intangible cultural heritage predominantly revolved around the formulation of design policies and methodologies. These design strategies and applications have substantially informed the creation of "culture-oriented products [8]" This inquiry deconstructs the distinctive regional attributes inherent in Chinese woodblock New Year pictures. Subsequently, it reconstructs the process of crafting derivatives utilizing the pertinent generative design frameworks pioneered by the Stanford University School of Design. To empirically investigate this paradigm, this study engages junior students specializing in product design as its subjects. The research unfolds as follows: Primarily, employing descriptive statistical analyses, the feasibility of generative Design across distinct non-legacy projects is scrutinized. Subsequently, generative design theory and user journey mapping are harnessed to invigorate innovative cogitation among students. Concluding this phase, derivative concepts are materialized through proposition design, followed by the application of dependent sample T-test to scrutinize experimental outcomes. Ultimately, it is observed that immersion in generative design practices enhances both the volume and caliber of design propositions. From a sensorial stimulation perspective, the study amalgamates findings to distill a design methodology, elucidating a generative Design approach that traverses from "vision and action to experiential facets." The conclusions drawn from this research introduce novel avenues for the perpetuation of intangible cultural heritage craftsmanship and augment the array of generative Design methodologies.

Transforming traditional intangible cultural heritage into modern derivative products through commercialization is an important means of preservation. Among these, the aesthetic appeal of the derivatives is a crucial factor. American economist Daniel Hamermesh, in his paper "Beauty and the Labor Market," pointed out that there is both an "Ugliness Penalty" and a "Beauty Premium" in society, indicating that physical attractiveness is positively correlated with overall labor income [9]. This preference for beauty extends to individuals, cultural artifacts, and commercial products [10].

The term "Beauty Economy" refers to the phenomenon where, in the context of product homogenization, users are more inclined to purchase products that offer aesthetic appeal and a superior user experience [11] This paper introduces this concept into the protection of Intangible Cultural Heritage, aiming to increase young people's interest in traditional culture through product innovation and to enhance the commercialization of Intangible Cultural Heritage through the power of design [12].

In contemporary times, propelled by the dynamic growth of the "Beauty economy" economy, experiential design, and the cultural and creative sectors, fostering "design innovation" for intangible cultural heritage has emerged as a prevailing commercial trajectory, giving rise to a systematically structured creative domain. The regional ethos encapsulated within intangible cultural heritage contributes a wealth of experiential dimensions to the "design innovation" sector, thereby bolstering its resonance within the global marketplace [ [ [13,14]. This inquiry centers on the symbiotic influence of intangible cultural heritage on the "creative design" domain, investigating how the former empowers the latter while engendering novel founts of creative inspiration for designers.

Against the backdrop of globalization, regional innovation assumes a distinctive mantle in shaping cultural symbolism and propelling economic advancement. As the quintessence of regional ethos, intangible cultural heritage assumes the mantle of continuity, dissemination, and evolution. On one facet, within the sphere of inheriting intangible cultural heritage skills, the foundation of cultural prosperity rests upon industrial vitality. Simultaneously, cultural value augmentation serves as the impetus for economic value augmentation [15]. Consequently, stakeholders within the creative industries must duly recognize their role in exporting regional culture to the global market, embracing the responsibility of cultivating a global cultural footprint.

Chinese woodcut New Year pictures exemplify quintessential instances of generative Design, distinguished by their inherent qualities of mass production and industrialization. These attributes furnish substantial advantages for the realm of derivative innovation:

(1) Precision Assembly Line Operation: Operating in meticulous accordance with its technological progression, the distinct tasks encompassing painting, engraving, printing, and color registration are meticulously subdivided.

(2) Standardization and Batch Fabrication: The exigency of satiating the collective desire to beseech auspiciousness engenders a pronounced demand for woodblock New Year pictures, necessitating production in batches to align with prevailing market requisites.

(3) Pronounced Regional Distinctiveness: The art of printing and the utilization of a monochromatic palette lend an element of individuality, underscored by the hallmark regional characteristics.

(4) Robust Communicative Foundation: With Minnan individuals diffusing the footprints of woodblock New Year pictures across the global sphere, these cultural emblems resonate more profoundly within the market's collective consciousness.

The research's architectural framework is visually depicted in Fig 1.

The primary objective of this research is to explore the feasibility and innovativeness of generative design in the protection and inheritance of Intangible Cultural Heritage. By employing literature review and experimental methods, this study investigates the advantages of generative design in terms of the quantity and quality of design proposals. From the perspective of design thinking, three derivative product design models are proposed, further enriching the methods of Intangible Cultural Heritage protection and pathways for its inheritance.

The innovations of this research are as follows: (1) Construction of a systematic design method. A new model for derivative product design is proposed, utilizing a method capable of generating diverse product design proposals in a short time; (2) Exhibition of diversified inheritance pathways. Various design carriers and methods are explored, including product design, digital design, and cultural innovation, enriching the design techniques and forms of Zhangzhou woodblock New Year picture derivatives; (3) Balancing innovative design with traditional inheritance. Through generative design, the works retain the characteristics of intangible cultural heritage while meeting modern aesthetic and practical needs. In summary, this research, from an industrial design perspective, explores the extraction and analysis of traditional elements and the integration of modern design methods to achieve a balance between traditional culture and contemporary design.

## 2. Literature review

### 2.1. Research status of Chinese intangible cultural heritage

The Chinese government has been protecting religious and related intangible cultural heritage through various measures such as the intangible cultural heritage protection list, intangible

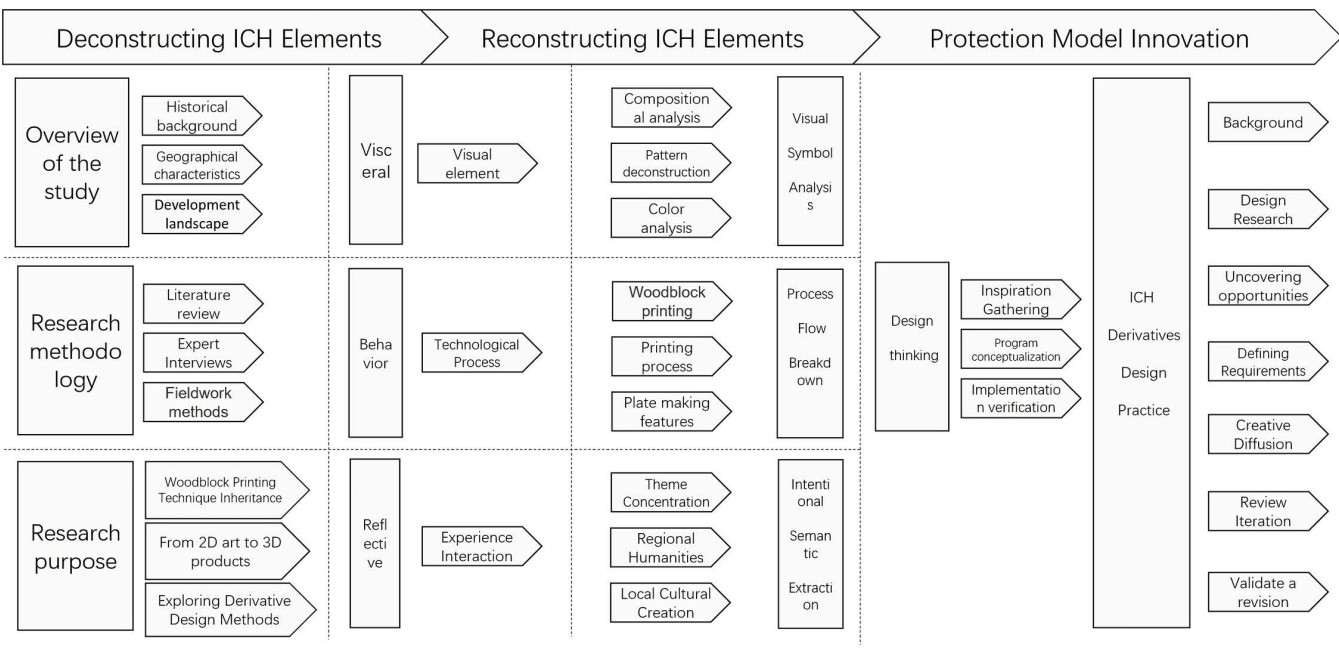

**Fig 1. Research Architecture Diagram.**

heritage inheritors, and cultural innovation [16]. Notably, in 2006, the first batch of the intangible cultural heritage list was announced, and protection has been achieved through recognition mechanisms, talent cultivation, and demonstration bases [1]. These efforts have yielded both macro and micro-level results, as detailed below:

### 2.1.1. Macro measures and outcomes.

(1) Policy Formulation: Enhancing intangible cultural heritage protection through legal regulations, thereby promoting cultural prosperity.

(2) Economic Benefits:Intangible cultural heritage contributes to the tourism economy, commodity economy, and city branding.

(3) Social Benefits:The protection and inheritance of intangible cultural heritage help reduce poverty and increase employment opportunities.

(4) Government Support for Regional Culture Promotion:Various religious groups have expanded their influence by showcasing, documenting, and promoting religious-related intangible cultural heritage [16].

These measures implicitly reflect innovative protection models and achievements, providing effective support for enriching the design of intangible cultural heritage derivatives.

### 2.1.2. Micro measures and outcomes.
The Chinese government has numerous successful experiences in policy formulation and implementation:

(1) Productive Protection Model: Integrating cultural and economic development goals for mutual reinforcement.

(2) Multi-level Legislative Framework: Legal guarantees for intangible heritage protection through comprehensive regional policies.

(3) Collaboration between Government and Academia: Actively supporting scholars in identifying, documenting, and theorizing intangible cultural heritage information.

(4) Active Use of New Media Creation Platforms: Diversifying promotional channels and methods to enhance the influence and public participation in intangible cultural heritage protection.

Overall, these innovative measures demonstrate China's comprehensive and diversified strategies for protecting intangible cultural heritage, encompassing legal protection, economic support, social participation, and new media dissemination [1].

Despite significant achievements in the protection policies and inheritance approaches for intangible cultural heritage, there are still issues such as limited protection methods, homogeneous inheritance carriers, and outdated design innovation methods. This study will explore derivative design methods and applications for intangible cultural heritage from the perspective of design innovation, further enriching the methods and product carriers for heritage inheritance.

The essence and transmission of Chinese traditional culture, as captured within non-legacy documentation, accentuate the concepts of "observing individuals, perceiving artifacts, and embracing existence." This implies that the skillful transmission by non-genetic heirs, the dissemination via non-legacy bearers, and the manifestation of non-legacy initiatives should seamlessly integrate into daily lifestyles. The UNESCO-formulated Convention for the Safeguarding of Intangible Cultural Heritage underscores that such heritage possesses the capacity to adapt to evolving contexts while maintaining continuity across generations. This adaptability is intertwined with its ongoing interaction with history, leading to constant reimagining within the framework of the times. Generative Design assumes a constructive role in safeguarding and perpetuating intangible cultural heritage.

## 2.2. Research status of skill inheritance of chinese woodcut new year pictures

The origins of Chinese woodcut New Year pictures can be traced back to the Song Dynasty, and it reached its zenith during the Ming and Qing Dynasties. This art form is hallmarked by its expansive compositions, diverse gradations of thickness, the harmonization of resilience and pliancy, distinct thematic content, and its resonance with the experiential aspirations of common folk within various life scenarios. Possessing a wide-reaching popularity and ubiquitous regional traits throughout the nation, notable examples include Yangliuqing New Year pictures, Wuqiang New Year pictures, Zhuxian Town New Year pictures, and Mianzhu New Year pictures. Over a span of millennia, the evolution of Chinese woodblock New Year pictures has culminated in the development of distinct regional attributes, a convergence brought forth by geographical attributes, cultural practices, and the essence of artisanal prowess. The subsequent exposition delineates its unique craftsmanship from two vantage points: material selection, plate crafting, and printing techniques

**2.2.1. Material selection and plate making.** Pear wood is the preferred material for woodblock carving due to its refined grain, facilitating a smooth carving process, and yielding a distinct print outcome. Among the favored paper choices are Minxi jade buckle paper, Wannian red paper, and bespoke tinted paper. In tandem with natural botanical and mineral pigments, the distinctive hallmark lies in the utilization of "white stain" (referred to as "big mold powder"). This substance results from grinding ore powder (specifically, white ridge soil) and amalgamating it with adhesive.

The engraving phase encompasses the "yin" and "yang" versions:

(1) The "yin" rendition pertains to the negative version of characters such as the "Young God," where the background color (typically red) is printed. To regulate moisture during edge printing, the engraving lines and the boundary of the color block exhibit an outward incline. The printing methodology involves superimposition: initial printing of color blocks followed by the application of black strokes for lines.

(2) The "yang" iteration delineates patterns through a strategic arrangement of dots, lines, and surfaces. Here, the engraving process encompasses crafting the background while preserving the central motif.

**2.2.2. Printing skills.** Conventional woodblock New Year picture production typically follows the sequence of printing black-line templates followed by color plates. Conversely, the technique employed in crafting Chinese woodblock New Year pictures involves a combination of printing, color separation overprinting, and watermark powder printing. Originating during the Ming Dynasty, this approach encompasses a woodblock watermarking system. The design is replicated onto several small wooden blocks, affixed in designated positions, and then sequentially superimposed with colors ranging from light to dark. This technique yields richly hued and multi-leveled printed outputs, yet the production process is intricate and intricate, akin to the accumulation of vibrantly hued confections resembling "bucket nails" within a vessel. Hence, the term "printing" was coined.

The process unfolds as follows: Begin by printing color plates in shades of green, white, red, and yellow, followed by the application of an ink draft. Its technical merits are two-fold: first, the line draft's alignment with the initial color draft is enhanced, minimizing white space and bolstering efficiency; second, it counteracts the potential of potent opacifying pigments to obscure the ink draft beneath.

From an academic standpoint, Chinese woodcut New Year pictures undergo diverse analytical lenses: exploring its image narratology [6], dissecting its design element composition through a semiotic prism [17], and investigating its characteristics from an artistic cross-disciplinary perspective [18]. To comprehensively grasp the regional "Woodblock New Year Pictures" phenomenon, the utilization of Vosviewer facilitates statistical analysis of literature's focal research domains.

VOSviewer can visualize the research history and hotspots in this field, providing direction for research trends and focal points. VOSviewer, developed by the Centre for Science and Technology Studies at Leiden University, is an open-source software tool that uses text mining of titles, authors, and keywords for term frequency, co-occurrence, and co-citation analysis. It aids researchers in visualizing bibliometric networks, commonly used to construct and visualize bibliometric maps within scientific domains, thus presenting research history and trends [19]. Its main functionalities include: (1) constructing co-occurrence, citation, and co-citation maps; (2) visualizing bibliometric networks; (3) text mining of literature to uncover research hotspots. Overall, VOSviewer is widely used in research evaluation, analysis of academic frontiers, and the construction of knowledge maps, providing robust support for understanding research backgrounds and trends [20]. This study employs VOSviewer for literature mining and utilizes visual maps to illustrate historical documents and research trends.

Commencing with "Woodblock New Year Pictures" as the search motif, the focal terms from published literature are meticulously harvested to facilitate word frequency analyses. Ultimately, a co-occurrence network knowledge map concerning "Woodblock New Year Pictures" is derived. The operational sequence unfolds as follows:

(1) Precise Keyword Retrieval: "Woodblock New Year Pictures" keywords are precisely queried within the CNKI of HowNet database, yielding a total of 2,876 search results, of which 720 are retrievable references.

(2) Visualization via VOSviewer: Employing the visualization software VOSviewer, a meticulous dissection of keyword frequency within the literature transpires. This culminates in the creation of a co-occurrence network that highlights the interrelations among keywords within China's research purview concerning "Woodblock New Year Pictures."

For a visual representation of this knowledge map, kindly refer to Fig 2.

The illustration distinctly depicts that previous academic discourse surrounding "woodblock New Year pictures" has primarily gravitated towards themes encompassing "folk art and traditional customs," "cultural innovation and creativity," and "intangible cultural heritage." An overarching trend is the recurrent exploration and application of regional attributes, exemplified by the study of "woodblock New Year pictures" in locales like Zhuxian Town, Dongchangfu, Taohuawu, Fengxiang, Laohekou, and Yangjiabu. Conversely, the investigation into "woodblock New Year pictures" within the Chinese area remains relatively sparse. This

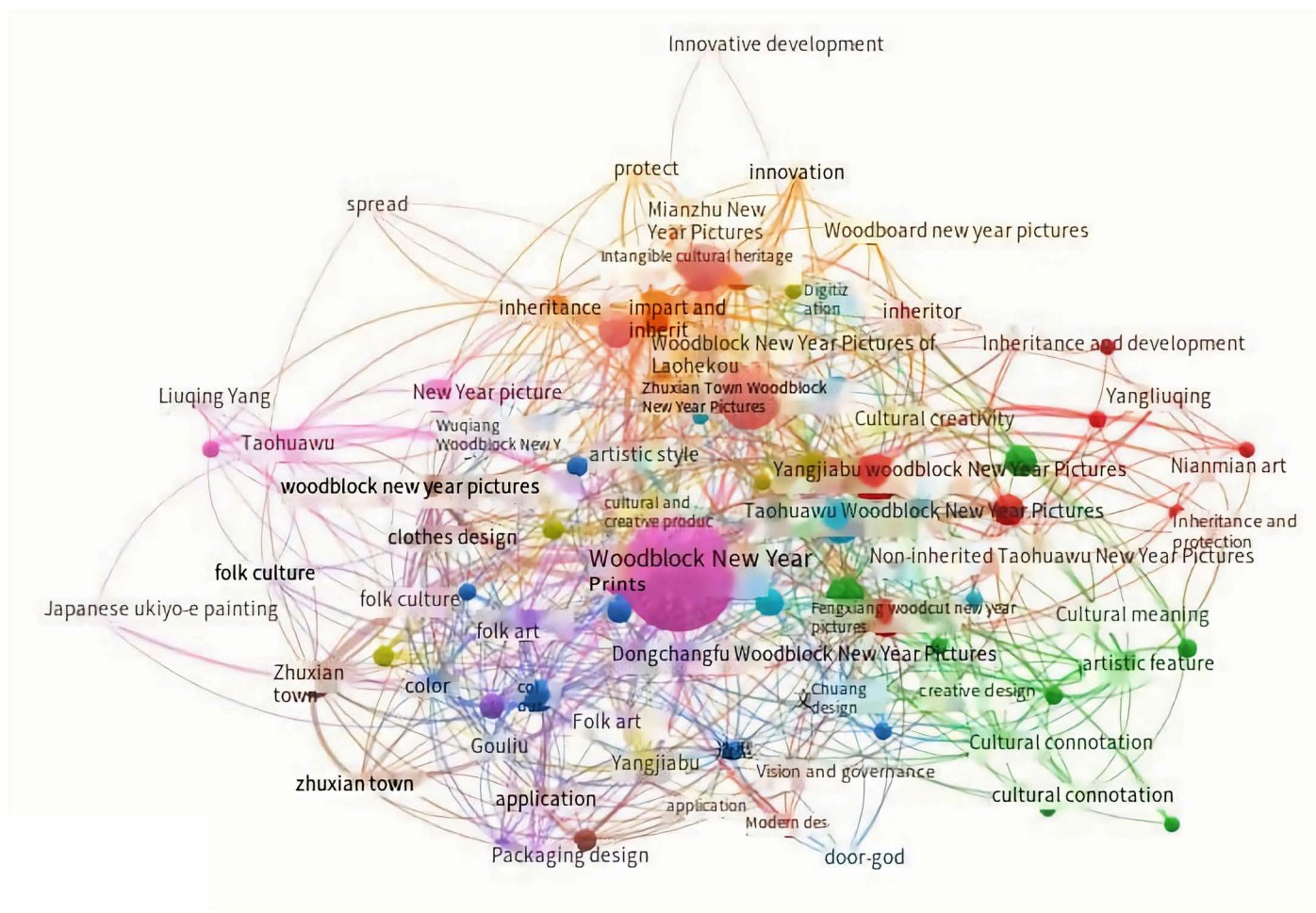

**Fig 2. Keyword co-occurrence network under the theme of "Woodcut New Year Picture".**

void heightens our inquisitiveness regarding the unfolding trajectory of research concerning "woodblock New Year pictures" within the Chinese vicinity.

## 2.3.  Research trend analysis of chinese woodblock new year pictures

Centered around the motif of "Chinese Woodcut New Year Pictures," a meticulous inquiry within the CNKI database yields 83 discernible search outcomes. Subsequently, the analytical capabilities of the VOSviewer visualization software are harnessed to meticulously inspect keyword frequency across the corpus of literature. This endeavor culminates in the construction of a co-occurrence network spotlighting the interconnectedness of keywords within the domain of research concerning Chinese Woodcut New Year Pictures within China. Kindly refer to Fig 3 for a visual representation of this knowledge map.

The visual representation conspicuously underscores that previous scholarly investigations into Chinese Woodcut New Year Pictures were predominantly centered around themes such as "regional innovation," "safeguarding of traditional craftsmanship," and "folk art." The prevailing discourse primarily revolved around the exigency of protection, often explored from the vantage points of policy and cultural dimensions. However, there remains a pronounced dearth of comprehensive deliberation on the specific methodologies and measures for safeguarding. In essence, the outcomes of this literature analysis highlight the profound concern of domestic scholars for the preservation and evolution of "Chinese Woodcut New Year Pictures." Nevertheless, the dialogue surrounding generative Design methods and practices in this context remains conspicuously limited.

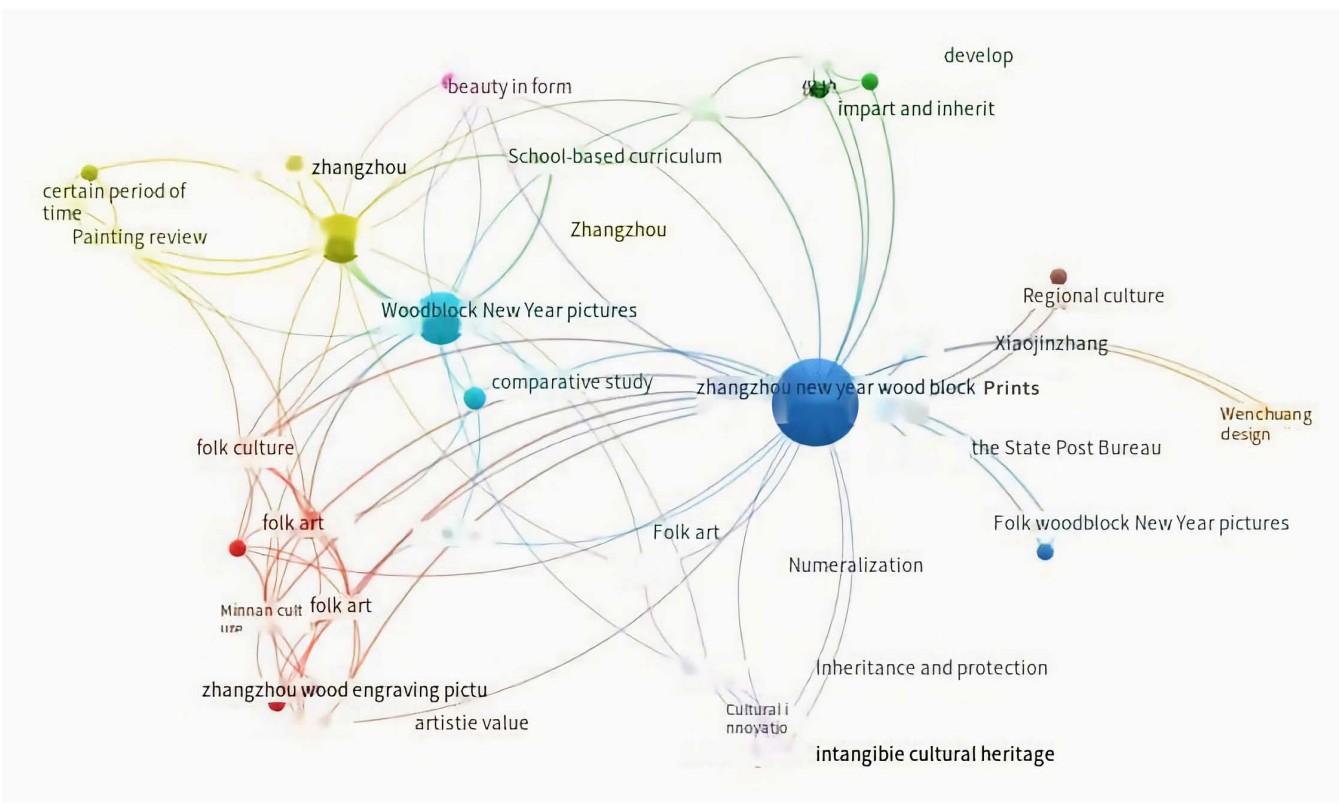

**Fig 3.  Keyword co-occurrence network under the theme of "Zhang Zhou Wooden New Year Pictures".**

## 2.4. Current status of generative design research

A derivative, in broad economic terms, signifies a novel creation stemming from the evolution of a foundational entity [21]. This concept finds particular resonance within the realm of financial instruments, where derivatives denote fresh investment products that emanate from specific assets. The English term "derivatives" refers to correlated entities arising from fundamental constituents, with its initial application rooted in the financial domain. In recent years, fueled by trends like cultural and creative design, surprise boxes, and the IP economy, generative Design has progressively entered the public consciousness. It pertains to innovations in application carriers, hinging on identical design elements. Non-legacy derivatives stem from intangible cultural heritage, encompassing both cultural and commercial attributes. Non-legacy generative Design constitutes an innovative endeavor that reinvents traditional culture [22].

Generative design, originally stemming from the field of computer science, is a design methodology that generates a multitude of design options based on predetermined rules and optimizes and filters them according to specific conditions [23]. In industrial design, it typically refers to the method where designers use modern design tools to extract traditional design elements and perform iterative design, systematic innovation, and derivative creation [24].

Generative design is widely applied in commercial design, engineering manufacturing, and the healthcare sector, characterized by its innovation, efficiency, low cost, automation, and scalability [25]. For instance, in product design, it assists designers in achieving intelligent design processes, enhancing the innovation and competitiveness of derivative products [26]. In engineering manufacturing, it accelerates product iteration and diversifies derivative products [27]; in architectural design, it provides new workflows and innovative methods for Building Information Modelling (BIM) [28]. In summary, generative design is an innovation-driven model based on rules, allowing designers to rapidly generate derivatives in various styles and categories. It can refresh the protection pathways of intangible cultural heritage and enrich the carriers of heritage transmission, possessing significant theoretical and practical value [29]

Generative design is a rule-driven innovation model widely used in fields such as architectural design, industrial design, engineering manufacturing, and commercial fashion. For instance, in industrial design, generative design can generate derivative product designs with different styles and aesthetic effects. Overall, generative design utilizes modern design tools to help designers break through traditional protection and transmission modes of intangible cultural heritage, exploring more diverse derivative design methods to achieve more efficient and innovative protection models [30].

**2.4.1. Generative design in intangible cultural heritage preservation.** Generative design facilitates the digital protection and diversified transmission of intangible cultural heritage, with advantages in the following aspects:

(1) Digital Protection and Transmission: Generative design first digitizes traditional elements of intangible cultural heritage, such as patterns, characteristic designs, and craftsmanship processes. Then, using algorithms or design rules, it generates numerous innovative design proposals based on these traditional elements, preserving and transmitting traditional culture in new forms.

(2) Efficiency and Low Cost:As an innovation model based on elements or rules, generative design can produce a large number of design proposals in a short time, reducing protection costs and enriching transmission carriers.

(3) Balance Between Inheritance and Innovation: Generative design retains cultural characteristics through key information extraction while satisfying modern cultural and economic needs with diverse derivative design proposals [31].

In summary, generative design addresses the challenges in traditional intangible cultural heritage protection, such as conservative cultural elements, outdated functions, and misalignment with modern consumer habits, by providing diversified transmission pathways and ensuring the continuity of cultural value.

By extracting traditional patterns, color compositions, craftsmanship, and cultural elements of Chinese woodblock New Year pictures, generative design facilitates innovative derivative product designs that meet the demands for personalization, diversity, and rapid iteration in the AI era. Its advantages are evident in the following aspects:

(1) Redesign of Traditional Culture:Deconstructing and reconstructing traditional elements to establish innovative principles, leading to diverse design proposals.

(2) Digital Dissemination and Protection:Using generative design to digitally archive Zhangzhou woodblock New Year pictures and generate various digital works, enhancing the spread and efficiency of intangible cultural heritage.

(3) Commercialization and Industrialization: Generative design offers rich traditional carriers and pathways, which, when combined with the cultural tourism industry, can enhance regional economy and influence.

By meticulously exploring the CNKI database with the thematic anchor of "Generative Design," 804 retrievable search outcomes surfaced. The application of the VOSviewer visualization software facilitated meticulous measurement and enumeration of the literature. This endeavor culminated in the formation of a co-occurrence network spotlighting keyword interconnections within the Chinese research arena of "generative Design." For a detailed visual portrayal, kindly refer to Fig 4.

The visual representation illustrates that past academic research trends in "generative Design" predominantly revolved around themes such as "brand design," "cultural and creative products," "intangible cultural heritage," "design strategy," "animation IP," and other related domains. While these research topics span a wide array of areas, encompassing art and folk traditions, there remains a paucity of comprehensive explorations into specific "generative Design" methodologies. Consequently, this scarcity impedes the direct provision of methodological guidance to practitioners within this sphere. As a response, this study undertakes an examination of the inheritance and preservation of "Chinese Woodcut New Year Pictures" through the lens of "generative Design." This approach seeks to propagate intangible cultural symbols by means of market dissemination and user engagement.

In conclusion, intangible cultural heritage urgently needs more systematic protection methods, richer product carriers, and diversified transmission pathways to meet the needs of the times.

**2.4.2. Generative design in woodblock new year pictures preservation.** Generative design's deconstruction and reconstruction of Zhangzhou woodblock New Year pictures' design elements not only enriches protection modes and transmission pathways but also imbues them with new vitality and value, aligning them with the economic and cultural demands of modern society [32].

Relative to traditional cultural design, generative Design possesses a broader spectrum of application, more robust communication mediums, and an expansive scope for research and development. Presently, research on generative Design encompasses aspects like modern

market integration from the technical viewpoints of parameterization and digitalization [33], dissecting its application pathways through the lens of cultural semantics [34], and investigating specific application cases such as furniture, animation, and tourism souvenirs [35]. The ensuing visual analysis data charts, rendered via the Vosviewer software, delineate its research focal points and evolving trends.

Generative design is an important means of protecting intangible cultural heritage. Within the realm of innovative techniques for non-legacy derivatives, diverse methodologies vie for prominence. Instances include constructing the generative Design prototype of "Tianjin Goubuli Steamed Bun" via extension semiotics [36], deconstructing the cultural elements of "Tujia brocade" through situational map methods [37], delineating the pattern composition principles of "Majiayao vortex pattern" with the assistance of Grasshopper parametric tools [12], and redefining the digital cultural creation of "Pingyang woodcut New Year pictures" through scene theory [13]. This study, however, endeavors to analyze the design process and methodologies of non-legacy derivatives through the prism of generative design. In essence, the research and development of non-legacy products serve as a potent avenue for invigorating non-legacy skills and ensuring the industrialization of non-legacy heritage [36].

The scope of generative Design exploration currently encompasses three dimensions:

(1) Embracing Doctrine and Formulating Reality: This involves re-engraving classics to reduce circulation costs. It revolves around utilizing art forms such as painting, sculpture, and cultural artifacts as objects for reproduction. For instance, the National Palace Museum in Taipei offers souvenirs like the carnival stone and jade cabbage [38].

(2) Transposing Flowers and Cultivating Trees: This technique involves grafting, extracting visual elements from reproduced objects such as form, patterns, and symbols. An

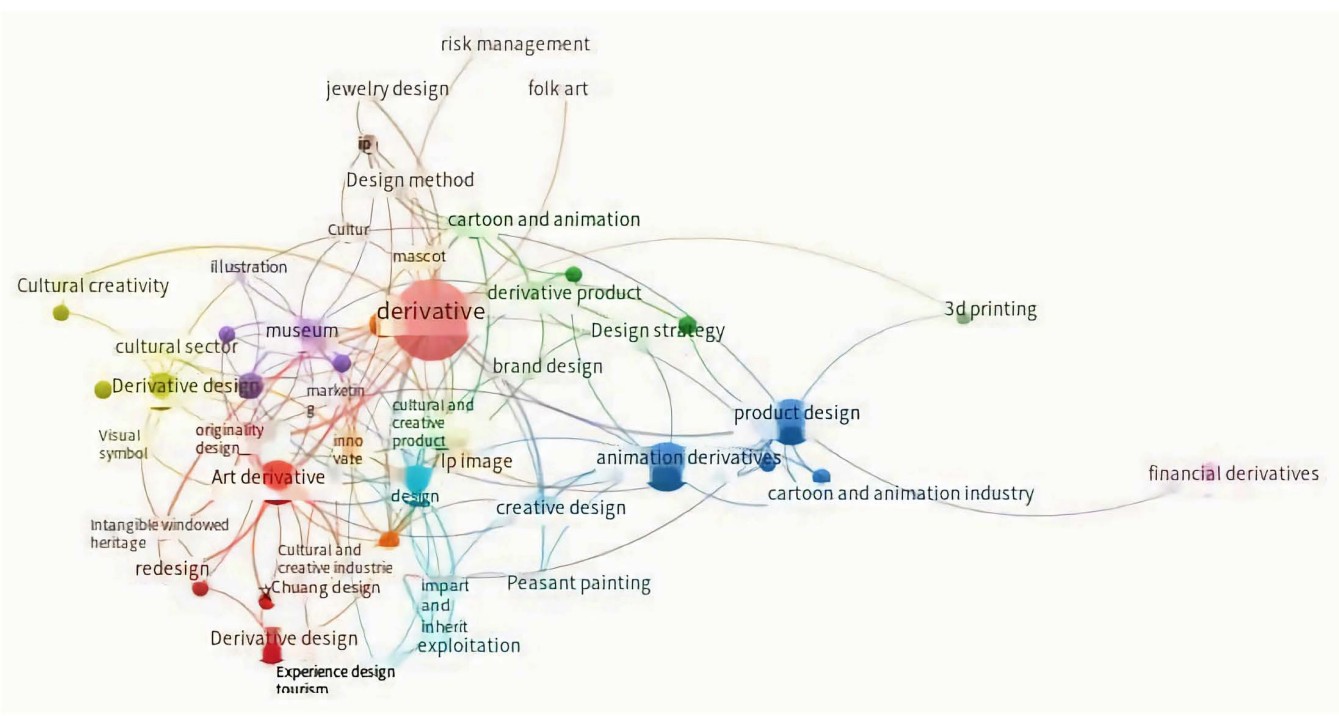

**Fig 4. Keyword co-occurrence network under the theme of " Generative Design ".**

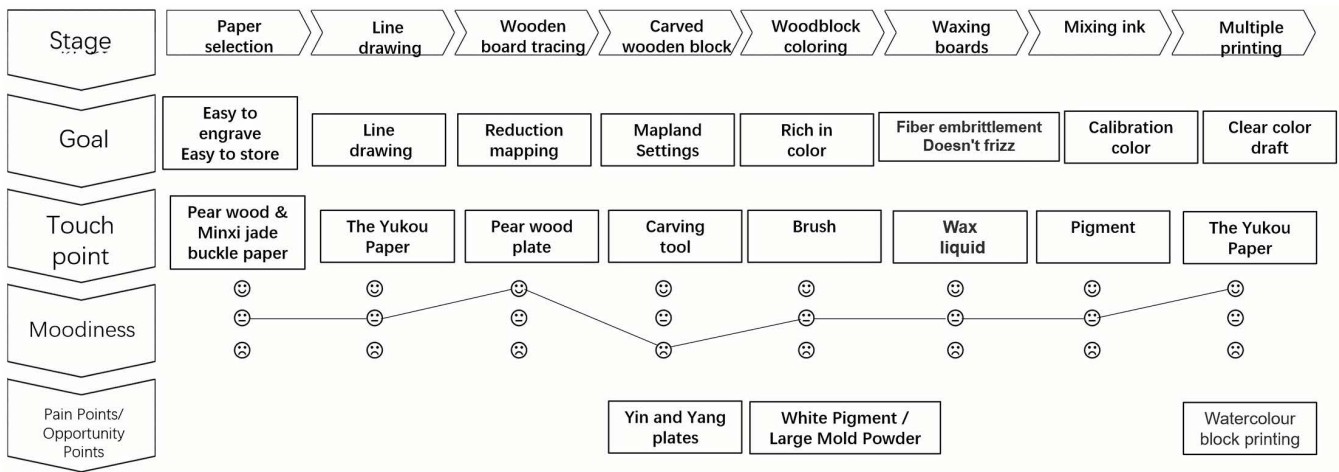

**Fig 5. The Journey Map of Chinese Woodblock New Year Paintings Craftsmanship.**

example is the "I Know" tape and the imperial edict notebook from the Palace Museum in Beijing [39].

(3)  Innovation through Transformation: This method involves deconstructing and reinterpreting original art forms. For instance, Littveld's Red and Blue Chair is a three-dimensional interpretation of Mondriaan's style of painting [40]. As illustrated in Fig 5.

## 3.  Research methods and processes

This study is structured into three distinct stages, with the overarching aim of comprehensively examining the noteworthy impact of generative design and tools on the creation of non-legacy derivatives. The stages are outlined as follows:

Stage 1: Exploration of Generative Design Potential
In the initial stage, the primary objective is to discern whether diverse categories of non-legacy projects offer viable avenues for generative Design.

Stage 2: Theoretical Instruction
The subsequent phase involves providing theoretical instruction to the experimental group. This instructional phase encompasses disseminating knowledge about the five-step generative design methodology, alongside acquainting participants with the operational intricacies of constructing a user journey map.

Stage 3: Investigation and Research
The third and final stage entails investigation and research endeavors. This involves proposition design, leading to the collection of two distinct groups of design sketches. Subsequently, an independent sample T-test is conducted to evaluate the statistical significance of differences between these groups.

A total of 46 participants took part in the experiment, falling within the age range of 20 to 22 years, with 65% of them identifying as female. These participants are third-year undergraduate students pursuing a major in product design. Each individual possesses a foundational grounding in design, having been trained in fundamental design processes, methodologies, presentation techniques, and courses focused on cultural creative design. Furthermore, the participants were divided into two groups, each consisting of 23 designers, utilizing a randomized allocation.

Table 1. Basic information of interviewee (N = 46).

| Profile | Items | Number | Percentage(%) |
|---|---|---|---|
| Gender | Male | 16 | 35% |
|  | Female | 30 | 65% |
| Age | 20 | 6 | 13% |
|  | 21 | 32 | 70% |
|  | 22 | 8 | 17% |
| Education | Junior | 46 | 100% |
| Major | Product design | 46 | 100% |
| Design Experience | Yes | 46 | 100% |

To ensure the stability of the research, the control group and experimental group remained consistent across three stages of experiments, allowing for a scientific comparison of the facilitative effects of derivative design methods on the redesign of intangible cultural heritage. All participants have completed courses in "Traditional Cultural Redesign" and "Cultural Creative Design," enabling them to proficiently utilize design software and manual sketching for design expression. The basic information of the subjects is presented in Table 1.

## 3.1. Experimental design process

Phase One: Cultural Creative Product Design Inspired by Different Intangible Cultural Heritage

Due to the diverse types of intangible cultural heritage, there are significant variations in their forms and design approaches. This study selects "Quanzhou Nanyin" representing auditory stimulation, "Zhangzhou Woodblock New Year Pictures" representing visual stimulation, and "Ming-style furniture craftsmanship" representing tactile stimulation as sample stimuli from a sensory perspective. Design students participating in the practice are tasked with innovatively designing derivative products based on these stimuli. Ultimately, it was found that the highest number of design proposals originated from the inspiration of "Zhangzhou Woodblock New Year Pictures" (visual stimuli). This suggests that visually inspired designs have an advantage in terms of the quantity of derivative products generated.

Phase Two: Training on Generative Design for the Experimental Group
This phase divides participants into an experimental group and a control group, each comprising 23 individuals. The experimental group undergoes systematic training in generative design methods, while the control group receives no training. Results indicate that designers trained in generative design can produce a greater quantity and higher quality of design proposals.

Phase Three: Cultural Creative Design Proposition Inspired by "Zhangzhou Woodblock New Year Pictures"

This phase aims to further explore the application value of generative design in design practice, focusing on deriving methods for designing products inspired by "Zhangzhou Woodblock New Year Pictures." Research results demonstrate that generative design can systematically propose derivatives for intangible cultural heritage, enriching modes of preservation and transmission. Detailed experimental data and results will be presented in the following sections.

In terms of the evaluation process, three judges independently assess all the collected sketches utilizing a five-point rating system. Subsequently, the average score derived from the judges' assessments is designated as the final score for each work. The judging panel comprises three experts: one is a senior designer with a decade of professional experience, another

serves as a university lecturer with four years of teaching tenure, and the third is a university lecturer with a twelve-year academic history.

## 3.2. Inspiration of non-legacy elements of different carrier types on generative design

Numerous non-legacy projects offer diverse potential for generative Design across various domains. In our experimentation, we adopted a non-legacy derivative approach rooted in the classification of sensory stimuli. This was undertaken to enhance participants' familiarity with non-legacy elements and to assess the feasibility of employing different non-legacy domains for generative Design. Grounded in the framework of five sensory stimuli, we strategically selected distinct forms of inspiration material. Specifically, we chose "Nanyin" as an auditory inspiration source, "Chinese Woodblock New Year Pictures" as a visual source of inspiration, and "Ming Furniture Making Skills" as a tactile source of inspiration for the experimentation process. For a comprehensive overview, please refer to Table 2.

The outcomes of the experiment reveal a notable trend: designers exhibit heightened creative enthusiasm when exposed to a more diverse array of inspirational materials, which in turn leads to an enhancement in the quality of their creations. Nonetheless, it's noteworthy that a deeper comprehension of more abstract or culturally rich heritage materials is essential to effectively engage with the inspiration. This aspect introduces a predicament, wherein the complexity of heritage materials can pose challenges to the creative process.

Consequently, a potential solution emerges: diversifying the modes of presenting intangible cultural heritage. By embracing varied approaches to convey heritage, the potential exists to not only amplify the quantity but also elevate the caliber of generative Design efforts.

## 3.3. The influence of generative design on the quantity and quality of non-legacy derivatives proposals

In order to validate the hypothesis that generative design contributes to enhancing both the quantity and quality of non-legacy derivative proposals, we conducted an exploratory experiment within a controlled environment. This experiment involved participants undertaking two specific tasks: firstly, extracting key design elements from the representative work "Lion Holding a Sword in Its Mouth" of Chinese woodcut New Year pictures, and subsequently, crafting generative Designs based on these elements. Notably, no restrictions were imposed on the specific categories of derivatives. Participants were actively encouraged to engage in brainstorming, generating a greater multitude of solutions. Eventually, they were prompted to select an idea for in-depth design, thus elevating the overall quality of the concepts

To further showcase the history, craftsmanship, cultural elements, and design features of Zhangzhou Woodblock New Year Pictures, this paper presents the cultural characteristics and historical evolution of woodblock prints through text, images, and videos. First, relevant information about its history and inheritance was extracted from the "China Intangible Cultural Heritage Network." It belongs to the category of traditional arts and crafts, with themes often covering historical stories, decorative patterns, and elements that convey wishes

**Table 2. The Fluence of Intangible Cultural Heritage Items with Different Sensory Stimuli on the Quantity and Quality of Generative Designs.**

| ICH Carrier | Sensory stimulation | Number of people | Number of derivative proposals | Number of derivative proposals |
|---|---|---|---|---|
| Nanyin | Auditory | 3 | 3 | 1 |
| Chinese Woodblock New Year Picture | Vision | 7 | 8 | 1.33 |
| Ming style furniture making techniques | Touch | 36 | 38 | 1.44 |

**Table 3. Analysis of Cultural Characteristics of "Chinese Woodblock New Year Pictures" with Different Themes.**

| Representative Works | Four Divine Beasts | The two spirits guarding the entrance to the house | The Mouse Wedding |
|---|---|---|---|
| Cultural Implications | Auspicious designs symbolizing the pursuit of happiness. Their creative elements are drawn from classic stories, historical culture, or words of blessing. | Designs for warding off evil. Their creative elements are derived from historical figures or religious symbols. | Decorative paintings showcasing local customs. Their creative elements stem from regional cultural characteristics and festive life scenes. |

for happiness [41]. Secondly, a video tutorial was used to demonstrate the craftsmanship and production process of Zhangzhou Woodblock New Year Pictures [42]. Finally, three representative works were selected for cultural and semantic interpretation. The following discussion, from the perspectives of "blessing," "warding off evil spirits," and "folk customs," highlights these cultural and artistic features through the selected works [43]. The specific representative works and descriptions of their cultural characteristics are shown in Table 3.

The experimental proposal involved the creation of derivatives inspired by the woodcut New Year pictures of "Lion Holding a Sword in Its Mouth." Importantly, there were no constraints on the selection of carriers, and participants were granted a time frame of 30 minutes for experimentation. To mitigate the influence of external variables, all participants were unfamiliar with the "Lion Holding a Sword in Its Mouth" woodcut New Year pictures prior to the experiment. This theme was deemed fitting for an experimental topic.

The experiment comprised two distinct groups of participants, each subjected to distinct training. The first group solely underwent the standard curriculum-based training provided by the school. Conversely, the second group was introduced to the tools associated with generative design flow and user journey mapping. In the outcome, the first group produced 30 sketch proposals, with an average score of 0.795 for each project. Meanwhile, the second group generated 38 sketch proposals, attaining an average score of 2.13 for each task. Please refer to Table 4 for a comprehensive overview.

In this experiment, the analysis employed an independent sample T-test to assess the impact of generative design on the quality of "non-legacy derivatives" proposals. The outcomes of the variance homogeneity test indicated a p-value of 0.137, which is greater than the threshold significance level of 0.05. Moreover, significant outliers were observed, and the distribution within each group exhibited a tendency toward normality.The T-test results are as follows: $t(44) = -3.309$, $p = 0.002$, $d = 0.91$. These results signify a pronounced difference arising from the influence of generative design training on the quality of the final design proposals. To elaborate, there are evident variations in the extent of improvement in design proposals between those who didn't receive generative design training ($M = 1.22$, $SD = 0.795$) and those who did ($M = 2.13$, $SD = 1.058$).

In summation, the integration of generative design principles and relevant tools is proven to empower designers to augment both the quantity and quality of their design proposals.

## 4. Conclusions

Empirical research was conducted to ascertain the potential of generative design in enhancing the creative realm of derivatives stemming from Chinese woodblock New Year pictures. The

**Table 4. T-test of generative design on the quality of intangible cultural heritage derivatives proposals.**

| | Average (standard deviation) | | Freedom | t-values | p-values | Effect quantity (D) |
|---|---|---|---|---|---|---|
| | Untrained generative design (N=23) | Generative design trained (N=23) | | | | |
| Extroversion | 1.22 (0.795) | 2.13 (1.058) | 44 | -3.309 | 0.002 | 0.91 |

findings substantiate that designers equipped with generative design training and adeptness in using user journey maps are capable of generating a greater number of design sketches marked by superior quality. Additionally, the evaluation scores assigned to each of their works are also notably elevated.

Concurrently, the process of deducing and dissecting the sketches has led to the formulation of a novel design concept. This concept is centered on the transformation of the design elements within non-legacy derivatives into derivative products. The intricacies of this concept are depicted in Table 3, providing a comprehensive overview of the evolved design approach. As illustrated in Fig 6.

Using Chinese woodcut New Year pictures as a prime illustration, we can substantially enhance the breadth of their inspirational resources by adopting the following approaches:

(1)  Elevating Visual Presentation Art: By deconstructing the traditional New Year pictures into distinct compositional forms, line draft sketches, and modular color components, we can augment their visual impact. This strategy serves to intensify the visual stimulation experienced by viewers.

(2)  Visualization of Production Process: Delving into the visualization of the woodblock New Year picture production process can yield a wealth of inspiration. Extracting intricate

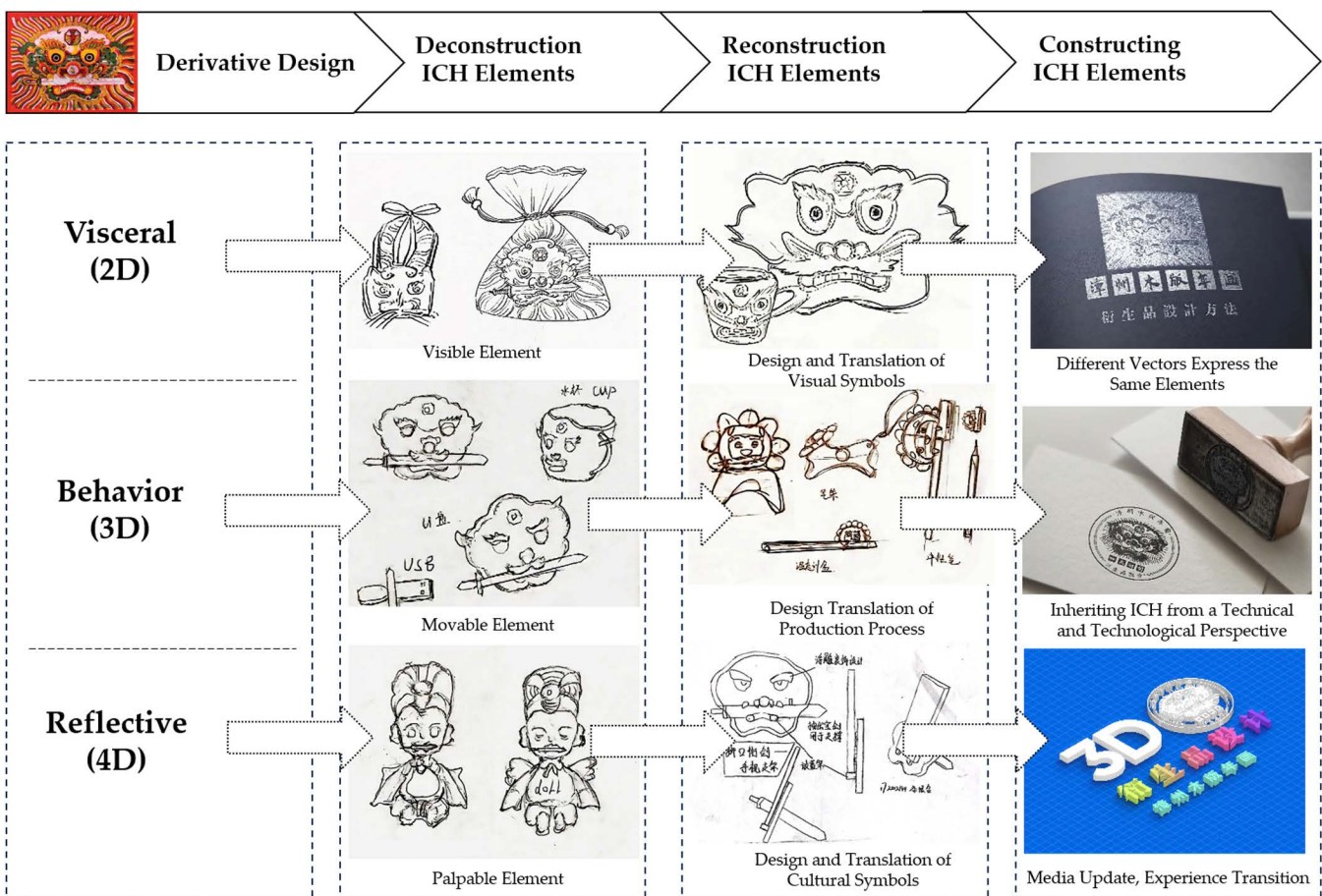

**Fig 6. Ideas for Construction and Translation of Derivatives of "Chinese Woodblock New Year Paintings".**

technological nuances involved in carving and plate creation, and then visually presenting them, offers designers an opportunity to transition from mere visual references to comprehensive behavioral action references.

(3) Interactive Design Through Experiential Scenes: Akin to an immersive experience, this perspective entails interpreting the interactive design of woodblock New Year pictures within contextual scenes. This approach not only incorporates innovative application methods and platforms but also delves into carriers. By engaging with the abstract cultural elements of yesteryears, like blessings, warding off malevolent forces, and traditional customs, designers are provided with a more fertile landscape for creativity to thrive. Refer to Fig 6 for a visual representation.

The amalgamation of these multifaceted perspectives leads to a holistic enrichment of the inspirational realm for Chinese woodcut New Year pictures, sparking innovative and diverse design directions. As illustrated in Fig 7.

An integral heuristic tool within the realm of generative design, the execution process of the user journey map can be dissected and effectively adapted as a reference process for non-legacy generative Design. In the context of Chinese woodcut New Year pictures, this process unfolds across three pivotal dimensions of sensory stimulation:

(1) Deconstruction of Visual Elements: Embarking on a deconstruction journey encompassing shape, color, material, and the intricate production process inherent in intangible carriers. By meticulously extracting these visual elements, they can subsequently be artfully integrated across diverse platforms to cater to the ever-evolving demands of the market. To illustrate, consider imprinting the "Lion Holding a Sword in Its Mouth" woodcut New Year picture pattern onto assorted cultural and creative carriers.

(2) Reconstruction of Interactive Scenes: Engage in the act of reimagining intangible interactive scenarios by extracting the essence of their production process, utilization modes, and atmospheric experiences. The insights thus garnered can be seamlessly applied to products sharing akin structures, functions, and operational sequences. For instance,

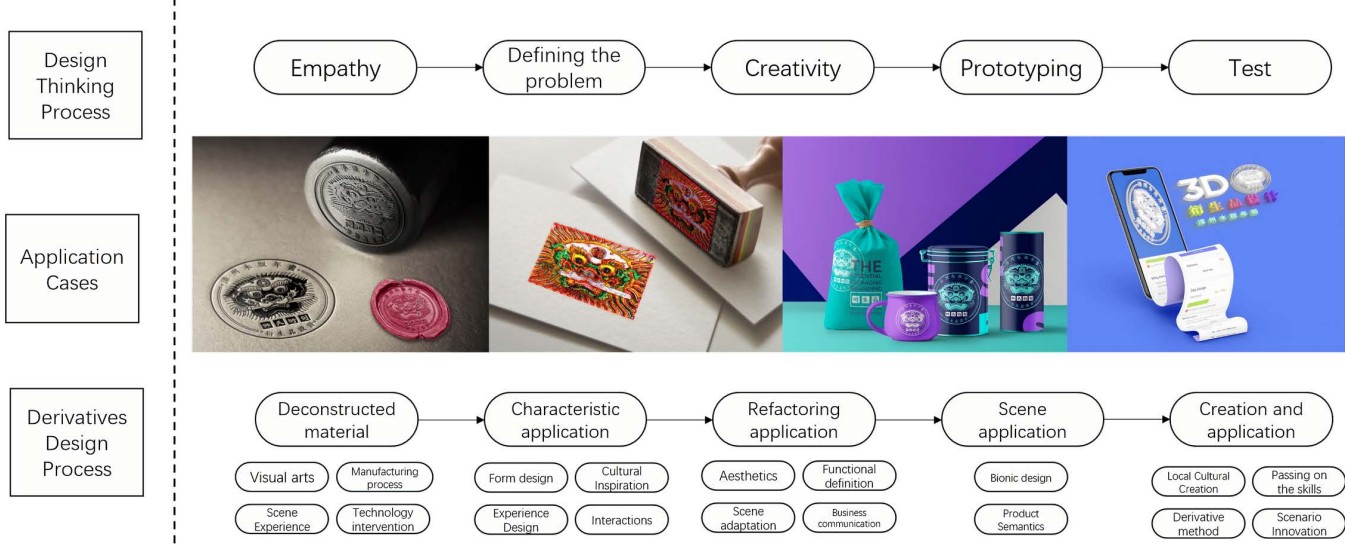

**Fig 7. Design method of derivatives of Chinese woodblock New Year pictures.**

transposing the mortise and tenon structure characteristic of Ming furniture-making skills onto structural entities such as building blocks and toys.

(3) Creation of Cultural Symbols: Champion the creation of intangible cultural symbols that resonate with the spirit encapsulated within intangible cultural heritage. By infusing novel functions, technologies, or interactive mediums, the conveyed essence of intangible cultural heritage can be exquisitely exported. A case in point is the adaptation of "Jingdezhen traditional blue-and-white porcelain making skills" into online apps or games, thereby facilitating the transmission of culture and immersive experiences.

Leveraging these strategic dimensions bestows the capacity to metamorphose intangible cultural heritage into concrete, captivating, and resonant derivatives that seamlessly align with contemporary sensitivities and cultural expressions. Empirical experiments underscore that the non-legacy generative Design methodology, when interwoven with the automatic layer of visual stimulation and the reflective layer of semantic inspiration, yields an augmented volume of design proposals coupled with elevated design caliber.

However, it's worth noting that the design effect elicited by the behavioral layer, catalyzed by the technological process, is somewhat less than optimal. The underlying cause lies in the fact that this particular layer necessitates a certain degree of foundational knowledge for its enlightenment. Consequently, there exists ample scope for its growth and development.

Given the distinct nature of the printing process intrinsic to Chinese woodblock New Year pictures, wherein color plates take precedence to ensure pictorial saturation, followed by the subsequent application of black line drafts, an opportunity emerges for extracting diverse printing methods through the lens of experience design. This approach engenders the innovation of derivatives, such as the creation of a "blind box seal," which is encapsulated in Fig 8.

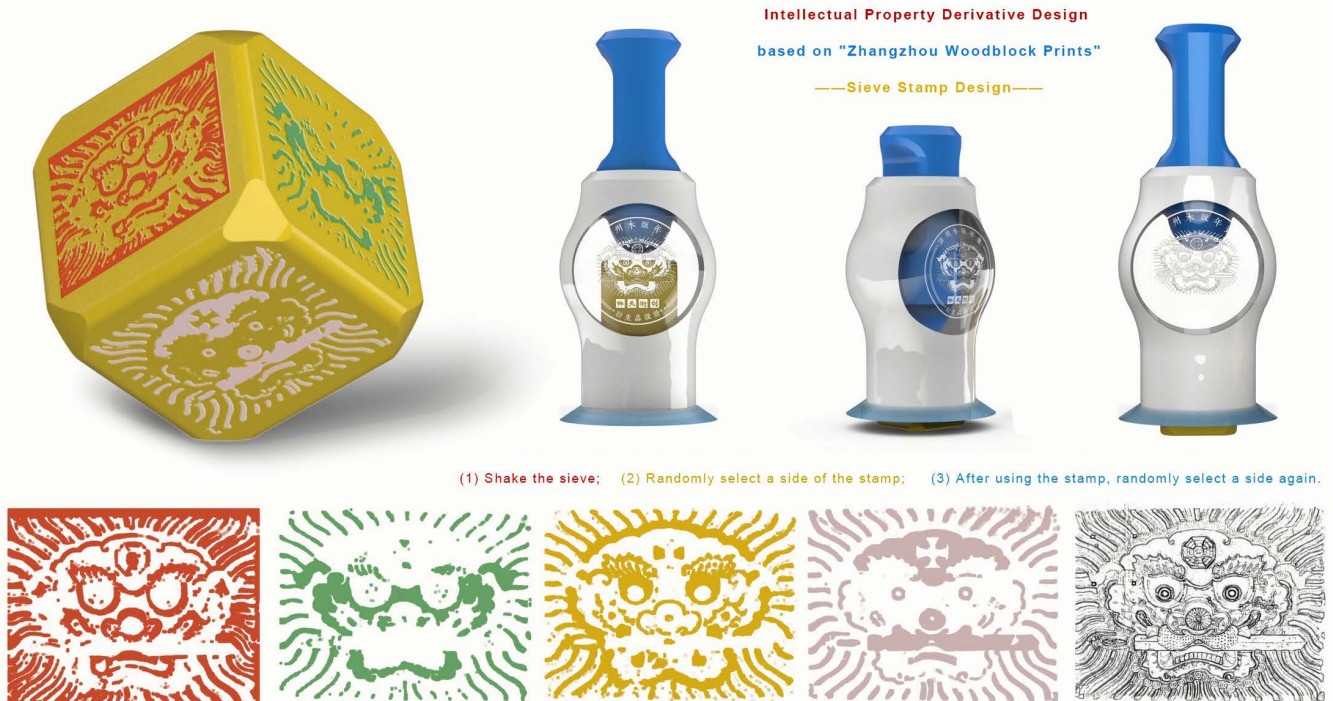

**Fig 8. The Application of Chinese Woodblock New Year Picture Elements in Blind Box Seal Design.**

This design is a manifestation of generative design, meticulously guided by the user journey map creation tool to navigate the course of generative Design. The intricacies of this approach are as follows:

(1) Automatic Layer Extraction: To initiate the process, the distinctive "Lion Holding a Sword in Its Mouth" pattern is judiciously extracted, a feat accomplished through the lens of the automatic layer.

(2) Behavior Layer Deconstruction: Progressing to the next phase, the overprinting mechanism intrinsic to Chinese woodblock New Year pictures is meticulously deconstructed, elucidating the behavioral layer.

(3) Reflection Layer Implementation: Drawing inspiration from the reflective layer, the differentiation of color plates materializes across six dimensions of a three-dimensional map. This is further augmented by the integration of elements such as the "twisted egg" and the widely embraced concept of a surprise box, endowing the seal with multifunctionality.

The act of sealing is rendered interactive, with users invited to engage in a random selection process by swinging. This serves a dual purpose: not only does it acquaint users with the overprinting procedure of woodblock New Year pictures, effectively capturing their attention and curiosity, but it also authentically mirrors the quintessential attributes of Chinese woodblock New Year pictures—specifically the printing process and application of white pigment.

Moreover, this design solution exudes universality, effortlessly harmonizing with the realm of three-dimensional creations sourced from two-dimensional art. In essence, it elevates traditional pattern symbols beyond mere visual representations, catalyzing their transformation into instruments that facilitate usage, drive behavioral shifts, and foster emotive engagement.

## 5. Discussion

The overarching aim of activating non-legacy lies in embracing the ethos of "observing people, observing things, and observing life." This pursuit entails not only the seamless transfer of non-genetic heritage but also its integration into the fabric of daily existence, invigorating it with vitality. The conduit for achieving this lies in generative Design, which rejuvenates the intangible cultural essence of spring snow, infusing it into the realms of sustenance, attire, abode, and conveyance for the Xialiba community. This approach conscientiously balances the imperatives of artistic inheritance with the dynamics of commercial circulation. A well-executed non-legacy generative Design endeavor holds tangible promise in fostering local cultural tourism, fortifying the social economy, and cultivating urban Intellectual Property.

Originating in southern Fujian and subsequently spanning across Hong Kong, Macao, Taiwan, and overseas domains, Chinese woodcut New Year pictures have transcended their initial folk context of prayers for blessings, protection from malevolent forces, and celebratory occasions. These pictures have organically evolved into a facet of China's profoundly rich historical and cultural legacy. As an emblem of regional culture, they weave a distinctive tapestry of aesthetic, behavioral, and experiential realms, endowing them with an inherent potential for crafting artistic derivatives.

This paper delves into the transformative significance of transitioning from "the preservation of intangible skills" to "the design of intangible derivatives." By dissecting the multifaceted connotations and extensions inherent to Chinese woodcut New Year pictures, a foundation is established for crafting derivatives that resonate with the intricate demands of the modern consumer milieu. Culminating this discourse, the paper introduces a generative Design paradigm that artfully interweaves two-dimensional and three-dimensional elements.

This revelation is intended to empower designers to harness their creative prowess in uncovering the latent strengths of traditional culture [20]. This study demonstrates the role of generative design in the protection of intangible cultural heritage. Through this approach, it is possible to generate numerous high-quality derivative product design proposals, providing new pathways for the transmission and preservation of intangible cultural heritage. In particular, the application of derivative design in the context of "Zhangzhou Woodblock New Year Pictures" has achieved innovative applications, transforming commercial products into digital art and evolving from two-dimensional to three-dimensional forms.

The experimental findings substantiate that the design quantity and quality of non-legacy derivatives can be enhanced through the application of generative design. However, it is important to acknowledge that this study has certain limitations, underscoring the potential for further exploration. The drawbacks encompass the following aspects: (1) The experiment exclusively revolves around the "lion with the sword" proposition, which may not be universally applicable across all non-legacy platforms; (2) The sample size gathered is relatively small, thus limiting the generalizability of the experimental outcomes; (3) The scope of the design method research is not exhaustive or sufficiently comprehensive, omitting other factors like the utilization of CMF (color, material, finish), prototype creation, etc.; (4) The study's focus is on novice designers, and the congruence of results with those of seasoned designers remains uncertain.

In future research endeavors, a dual-pronged approach can be pursued. Firstly, broadening the spectrum and depth of research subjects, incorporating a wider array of "intangible elements," and expanding the range of tested samples. Secondly, intensifying the number of research result iterations and refining the practicality of generative Design methodologies through an array of experimental findings.

## Supporting information

**S1 File. Summary of design sketches.**
(PDF)

**S2 File. Quantity and Score table of design schemes.**
(CSV)

## Acknowledgments

This work was supported by the 2023 Fujian Provincial Philosophy and Social Science Research Program for the Education System: Research on Digital Exploration and Application under the Integration of Religion and Cultural Tourism in Quanzhou (Grant No. JAS23008).

## Author contributions

**Writing – original draft:** Shao-Feng Wang.

**Writing – review & editing:** Chun-Ching Chen.

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
