## [Decision Letter · Decision Letter 0]

20 May 2024

PONE-D-24-04455Revitalizing Intangible Cultural Heritage via Derivative Design: A Focus on Chinese Woodblock PrintingPLOS ONE

Dear Dr. Wang,

Thank you for submitting your manuscript to PLOS ONE. After careful consideration, we feel that it has merit but does not fully meet PLOS ONE’s publication criteria as it currently stands. Therefore, we invite you to submit a revised version of the manuscript that addresses the points raised during the review process.

**ACADEMIC EDITOR: **

Dear authors

After reading the revisions provided by two independent experts on the topic, my suggestion for the authors is to revise and submit (major revisions) the manuscript by incorporating and addressing all the concerns indicated by the two reviewers. Please check and incorporate all the changes and modifications you consider in order to meet the satisfaction of the independent experts. In case it is required and additional expert could be additional review the manuscript.

With my best regards, the Academic Editor.

We look forward to receiving your revised manuscript.

Kind regards,

Jose Balsa-Barreiro

Academic Editor

PLOS ONE

Journal Requirements: ping

3. We note that your Data Availability Statement is currently as follows: [All data generated or analyzed in the course of this study are included in this published article and its supplementary file.]

4. PLOS requires an ORCID iD for the corresponding author in Editorial Manager on papers submitted after December 6th, 2016. Please ensure that you have an ORCID iD and that it is validated in Editorial Manager. To do this, go to ‘Update my Information’ (in the upper left-hand corner of the main menu), and click on the Fetch/Validate link next to the ORCID field. This will take you to the ORCID site and allow you to create a new iD or authenticate a pre-existing iD in Editorial Manager. Please see the following video for instructions on linking an ORCID iD to your Editorial Manager account: https://www.youtube.com/watch?v=_xcclfuvtxQ".

6. We note that Figure(s) 6, 7 and table 3 in your submission contain copyrighted images. All PLOS content is published under the Creative Commons Attribution License (CC BY 4.0), which means that the manuscript, images, and Supporting Information files will be freely available online, and any third party is permitted to access, download, copy, distribute, and use these materials in any way, even commercially, with proper attribution. For more information, see our copyright guidelines: http://journals.plos.org/plosone/s/licenses-and-copyright.

a. You may seek permission from the original copyright holder of Figure(s) 6, 7 and table 3 to publish the content specifically under the CC BY 4.0 license. 

Additional Editor Comments:

Dear authors,

After reading the revisions provided by two independent experts on the topic, my suggestion for the authors is to revise and submit (major revisions) the manuscript by incorporating and addressing all the concerns indicated by the two reviewers.

With my best regards, the Academic Editor.

Reviewers' comments:

Reviewer's Responses to Questions

**Comments to the Author**

1. Is the manuscript technically sound, and do the data support the conclusions?

Reviewer #1: Partly

Reviewer #2: Yes

2. Has the statistical analysis been performed appropriately and rigorously? 

Reviewer #1: I Don't Know

Reviewer #2: Yes

3. Have the authors made all data underlying the findings in their manuscript fully available?

Reviewer #1: No

Reviewer #2: Yes

4. Is the manuscript presented in an intelligible fashion and written in standard English?

Reviewer #1: Yes

Reviewer #2: Yes

5. Review Comments to the Author

Reviewer #1: The article presents a commendable scholarly attempt to navigate how innovation within derivative design fields may contribute to intangible cultural heritage safeguarding. The authors focus on Chinese woodblock New Year pictures, navigating transitioning from their traditional two-dimensional form to a more contemporary three-dimensional derivative design. By harnessing design thinking and a comprehensive experiential approach, the study unearths a novel compendium of derivative design methods, bridging the gap between traditional craftsmanship and modern design methodologies.

Whilst the paper harbours the potential to contribute significantly to the discourse on intangible cultural heritage and derivative design, there are several areas where substantial modifications could enhance its clarity, rigour, and overall impact.

The introduction, for instance, appears somewhat prolix, lacking a sharp articulation of the study’s primary objectives and intended contribution to the existing literature. A more focused and better-articulated introduction could better highlight the investigation’s novelty and its significance within the broader academic conversation.

The methodological exposition, particularly concerning the demographic characteristics of the control and experimental groups, demands a more detailed delineation. Clarifying these groups’ demographic traits, alongside a thorough discussion on whether the control and experimental groups were kept unchanged throughout the experimentation process, is imperative for ensuring the study’s robustness. Furthermore, the experimental setup, especially regarding participants’ awareness of their involvement in an experiment and the measures taken to mitigate the Hawthorne effect, warrants further explanation to fortify the study’s validity. Also, the final two colomns of Table 1 have the same heading. I suspect the right most heading is wrong. The authors should double check.

Another issue arises with the visual representation of literature through VOSViewer visualisation. While this approach is commendably innovative, a detailed introduction to the VOSViewer platform is necessary to ensure accessibility for all readers. Additionally, including a bilingual (Chinese-English) translation of the keywords would immensely aid in comprehending the thematic discoveries of the study, enhancing its international appeal and readability.

Moreover, specific terminologies and descriptions within the paper, such as the ‘“Yan value” economy’ (page 2, last paragraph, line 1), require further clarification. Also, is Figure 5, which depicts the Chinese woodcut New Year picture production process, original or adapted from other people’s research results? An appropriate citation from existing literature is needed if it is the latter.

Lastly, the discussion on safeguarding China’s intangible cultural heritage and its implications for economic activities and cultural innovation touches upon an area of great significance. It would, however, benefit from a more detailed engagement with authoritative literature in the field, especially regarding the influence of Chinese government policies on the creative economy within the intangible cultural heritage domain. More discussions on this topic, supported by relevant scholarly works, could enrich the paper’s contribution to understanding the socio-economic dimensions of intangible cultural heritage preservation and innovation. The authors may want to read and consider citing the following articles:

Xu, Y., Tao, Y., & Smith, B. (2022). China’s emerging legislative and policy framework for safeguarding intangible cultural heritage. International Journal of Cultural Policy, 28(5), 566-580. https://doi.org/10.1080/10286632.2021.1993838

Xu, Y., & Tao, Y. (2022). Religion-Related Intangible Cultural Heritage Safeguarding Practices and Initiatives of the Contemporary Chinese State. Religions, 13(8), [687]. https://doi.org/10.3390/rel13080687

In conclusion, this article is a pioneering investigation into the adaptive reuse and innovation of intangible cultural heritage within modern design methodologies. Hopefully, my recommendations will aid in refining the study.

Reviewer #2: In the introduction, it was difficult to follow along with what the authors were setting out to do with their research. I suggest they provide some simple definitions and briefly discuss the main point of the research paper more simplistically.

In the introduction, I suggest doing the following.

Provide a more simple definition of derivative design.

Then, discuss the significance of derivative design and its relationship with intangible heritage. I would even suggest starting the introduction with this main point and then moving into the background information on the Chinese woodblock New Year pictures so that the reader can more easily follow along and understand the significance of the study.

6. PLOS authors have the option to publish the peer review history of their article (what does this mean? ). If published, this will include your full peer review and any attached files.

**Do you want your identity to be public for this peer review?** For information about this choice, including consent withdrawal, please see our Privacy Policy .

Reviewer #1: No

Reviewer #2: No

---

## [Author Response · Author response to Decision Letter 0]

10 Jul 2024

Dear Editors and Reviewers:

Thank you for your letter and for the reviewers’comments concerning our manuentitled “Revitalizing Intangible Cultural Heritage via Derivative Design A Focus on Chinese Woodblock Printing”([PONE-D-24-04455] - [EMID:b43e276b40aaedd0]). Those comments are all valuable and very helpful for revising and improving our paper, as well as the important guiding significance to our researches. We have studied comments carefully and have made correction which we hope meet with approval. Revised portion are marked in red in the paper.

The main corrections in the paper and the responds to the reviewer’s comments are as flowing:

Responds to the Editor’s comments:

1. Response to comment: Please provide additional details regarding participant consent.

Response: This study is conducted as a routine part of regular teaching activities, with the experimental procedures forming an integral component of the curriculum. Oral consent has been obtained from all participants. The instructional content and processes have been ethically reviewed and approved by the ethics committee, obviating the need for additional documentation.

2. Response to comment: Data Availability Statement

Response: All data generated or analyzed in the course of this study are included in this published article and its supplementary file.

3. Response to comment: Additional corresponding author's ORCID iD

Response: Thank you very much for your reminder. I have added all the information.

4. Response to comment: Copyright confirmation of the image of Figure(s) 6, 7 and table 3.

Response: All images presented in this study are original creations by the author, including the contents of Figures 6 and 7, as well as Table 3 (revised as Table 4). As a design professional and educator, Figure 6 illustrates a synthesis of derivative design processes and original case studies. Figure 7 represents design practices conducted by the author based on the design methods proposed in this paper. The images in Table 3 (revised as Table 4) are also original creations by the author. There are no copyright issues involved. Thank you for your attention.

Thank you, editor, for your meticulous guidance and assistance. Your suggestions have been invaluable to me! In addition, this study added 21 new references. Notably, the content of the introduction, discussion, and experimental design sections has been expanded, adding a total of 2,300 words to the entire text. However, I am sorry that I cannot find the submission template of PLOS. I need your professional guidance and help in format adjustment.

Responds to the reviewer’s comments:

Reviewer #1:

1. Response to comment: The introduction lacks a clear statement of the main objectives of the study.

Response: Thank you to the reviewer for diligently revising this paper and providing detailed guidance on the author's revision process.

The primary objective of this research is to explore the feasibility and innovativeness of generative design in the protection and inheritance of Intangible Cultural Heritage. By employing literature review and experimental methods, this study investigates the advantages of generative design in terms of the quantity and quality of design proposals. From the perspective of design thinking, three derivative product design models are proposed, further enriching the methods of Intangible Cultural Heritage protection and pathways for its inheritance. （Line151-156）

2. Response to comment: The introduction lacks a clear articulation of the intended contribution to the existing literature.

Response:

2-1. Literature support for novelty

The practical application of Intangible Cultural Heritage in design innovation mostly remains at the stage of aesthetic form, lacking systematic innovation from the perspective of cultural elements and craftsmanship [3]. Current shortcomings in design innovation for Intangible Cultural Heritage protection are mainly reflected in the following aspects: (1) innovation only in form, without a profound understanding of the importance of cultural values[4]; (2) excessive pursuit of marketization and commercialization, leading to homogenization of existing products. While commercialization can expand the scope of dissemination, the single-minded pursuit of economic benefits weakens the original cultural and historical significance of Intangible Cultural Heritage [5]; (3) limitations of design innovation methods. The failure to adopt modern design thinking and tools leads to creative exhaustion[6]; (4) the limitations of dissemination media affect the range and acceptance of Intangible Cultural Heritage [7]。. In conclusion, to address the limitations of Intangible Cultural Heritage protection methods and dissemination channels, researchers should explore innovative thinking and methods from the root and apply them to specific cases for refinement and iteration. In the field of design, updating and applying design methods is the source of innovation. （Line42-56）

2-2. Innovation of this study

The innovations of this research are as follows: (1) Construction of a systematic design method. A new model for derivative product design is proposed, utilizing a method capable of generating diverse product design proposals in a short time; (2) Exhibition of diversified inheritance pathways. Various design carriers and methods are explored, including product design, digital design, and cultural innovation, enriching the design techniques and forms of Zhangzhou woodblock New Year picture derivatives; (3) Balancing innovative design with traditional inheritance. Through generative design, the works retain the characteristics of intangible cultural heritage while meeting modern aesthetic and practical needs. In summary, this research, from an industrial design perspective, explores the extraction and analysis of traditional elements and the integration of modern design methods to achieve a balance between traditional culture and contemporary design. （Line157-167）

3. Response to comment: More detailed description of the demographic characteristics of the control and experimental groups.

Response: To ensure the stability of the research, the control group and experimental group remained consistent across three stages of experiments, allowing for a scientific comparison of the facilitative effects of derivative design methods on the redesign of intangible cultural heritage. All participants have completed courses in "Traditional Cultural Redesign" and "Cultural Creative Design," enabling them to proficiently utilize design software and manual sketching for design expression. The basic information of the subjects is presented in Table 1. （Line471-476）

Table 1. Basic information of interviewee (N = 46)

Profile Items Number Percentage(%)

Gender Male 16 35%

Female 30 65%

Age 20 6 13%

21 32 70%

22 8 17%

Education Junior 46 100%

Major Product design 46 100%

Design Experience Yes 46 100%

4. Response to comment: Experimental setup, particularly regarding participants' awareness of their involvement and measures taken to mitigate the Hawthorne effect, need further explanation to enhance the study's validity.

Response:

Phase One: Cultural Creative Product Design Inspired by Different Intangible Cultural Heritage

Due to the diverse types of intangible cultural heritage, there are significant variations in their forms and design approaches. This study selects "Quanzhou Nanyin" representing auditory stimulation, "Zhangzhou Woodblock New Year Pictures" representing visual stimulation, and "Ming-style furniture craftsmanship" representing tactile stimulation as sample stimuli from a sensory perspective. Design students participating in the practice are tasked with innovatively designing derivative products based on these stimuli. Ultimately, it was found that the highest number of design proposals originated from the inspiration of "Zhangzhou Woodblock New Year Pictures" (visual stimuli). This suggests that visually inspired designs have an advantage in terms of the quantity of derivative products generated.

Phase Two: Training on Generative Design for the Experimental Group

This phase divides participants into an experimental group and a control group, each comprising 23 individuals. The experimental group undergoes systematic training in generative design methods, while the control group receives no training. Results indicate that designers trained in generative design can produce a greater quantity and higher quality of design pro-posals.

Phase Three: Cultural Creative Design Proposition Inspired by "Zhangzhou Woodblock New Year Pictures"

This phase aims to further explore the application value of generative design in design practice, focusing on deriving methods for designing products inspired by "Zhangzhou Wood-block New Year Pictures." Research results demonstrate that generative design can systematically propose derivatives for intangible cultural heritage, enriching modes of preservation and trans-mission. Detailed experimental data and results will be presented in the following sections. （Line482-504）

5. Response to comment: Furthermore, the last two columns in Table 1 have the same heading. I suspect the rightmost heading is incorrect. The author should review it carefully.

Response: Thank you for pointing out the error, which was due to inappropriate translation. After consulting with experts, the appropriate title for the rightmost column in Table 1 is "Protection model innovation." I appreciate the reviewer's feedback in identifying this oversight. （Line 446- Fig 5.）

6. Response to comment: It is necessary to provide a detailed introduction to the VOSViewer platform to ensure accessibility for all readers.

Response: VOSviewer can visualize the research history and hotspots in this field, providing direction for research trends and focal points. VOSviewer, developed by the Centre for Science and Tech-nology Studies at Leiden University, is an open-source software tool that uses text mining of titles, authors, and keywords for term frequency, co-occurrence, and co-citation analysis. It aids re-searchers in visualizing bibliometric networks, commonly used to construct and visualize bibli-ometric maps within scientific domains, thus presenting research history and trends [20]. Its main functionalities include: (1) constructing co-occurrence, citation, and co-citation maps; (2) visualizing bibliometric networks; (3) text mining of literature to uncover research hotspots. Overall, VOSviewer is widely used in research evaluation, analysis of academic frontiers, and the construction of knowledge maps, providing robust support for understanding research back-grounds and trends [21]. This study employs VOSviewer for literature mining and utilizes visual maps to illustrate historical documents and research trends. （Line266-277）

7. Response to comment: Furthermore, specific terms and descriptions in the paper, such as "Yan value" economy need further clarification.

Response: The English translation for " Yan value economy " should indeed be "Beauty Economy." I apologize for any confusion caused by my limited proficiency in English. Thank you for clarifying. Please provide the specific literature references and application descriptions for further assistance.

Transforming traditional intangible cultural heritage into modern derivative products through commercialization is an important means of preservation. Among these, the aesthetic appeal of the derivatives is a crucial factor. American economist Daniel Hamermesh, in his paper "Beauty and the Labor Market," pointed out that there is both an "Ugliness Penalty" and a "Beauty Premium" in society, indicating that physical attractiveness is positively correlated with overall labor income[10]. This preference for beauty extends to individuals, cultural artifacts, and commercial products [11].

The term "Beauty Economy" refers to the phenomenon where, in the context of product homogenization, users are more inclined to purchase products that offer aesthetic appeal and a superior user experience[12] This paper introduces this concept into the protection of Intangible Cultural Heritage, aiming to increase young people's interest in traditional culture through product innovation and to enhance the commercialization of Intangible Cultural Heritage through the power of design[13]. （Line102-114）

8. Response to comment: Was the process of creating woodblock New Year pictures in Figure 5 an original contribution or an adaptation of existing research? If the latter, appropriate citations are needed from existing literature.

Response: The process of creating woodblock New Year pictures in Figure 5 is original to the author. The author synthesized this craft process through field visits to museums in Zhangzhou, Fujian Province, interviews with intangible cultural heritage inheritors, literature review, and collection of videos and images. The production process of Zhangzhou woodblock New Year pictures is depicted in a concise and comprehensible process diagram. （Line 446- Fig 5.）

9. Response to comment:The discussion on the protection of Chinese intangible cultural heritage and its impact on economic activities and cultural innovation involves a critically significant field. However, further engagement with authoritative literature in this area is necessary.

Response: Firstly, I appreciate the careful guidance from the reviewer. The references you provided have deepened my understanding of China's policies and methods for safeguarding intangible cultural heritage. Through the literature review process, I thoroughly studied "China’s emerging legislative and policy framework for safeguarding intangible cultural heritage" and "Cultural impacts of state interventions: Traditional craftsmanship in China’s porcelain capital in the mid to late 20th century." Here is a detailed summary of China's supportive policies and the effectiveness of its efforts in protecting intangible cultural heritage:

2.1. Research status of Chinese intangible cultural heritage

The Chinese government has been protecting religious and related intangible cultural her-itage through various measures such as the intangible cultural heritage protection list, intangible heritage inheritors, and cultural innovation [17]. Notably, in 2006, the first batch of the intangible cultural heritage list was announced, and protection has been achieved through recognition mechanisms, talent cultivation, and demonstration bases[1]. These efforts have yielded both macro and micro-level results, as detailed below:

2.1.1. Macro Measures and Outcomes:

（1） Policy Formulation:Enhancing intangible cultural heritage protection through legal reg-ulations, thereby promoting cultural prosperity.

（2） Economic Benefits:Intangible cultural heritage contributes to the tourism economy, commodity economy, and city branding.

（3）Social Benefits:The protection and inheritance of intangible cultural heritage help reduce poverty and increase employment opportunities.

（4）Government Support for Regional Culture Promotion:Various religious groups have ex-panded their influence by showcasing, documenting, and promoting religious-related intangible cultural heritage [17].

These measures implicitly reflect innovative protection models and achievements, providing effective support for enriching the design of intangible cultural heritage derivatives. （Line169-188）

2.1.2. Micro Measures and Outcomes:

The Chinese government has numerous successful experiences in policy formulation and implementation:

（1）Productive Protection Model: Integrating cultural and economic development goals for mutual reinforcement.

（2） Multi-level Legislative Framework: Legal guarantees for intangible heritage protection through comprehensive regional policies.

（3） Collaboration between Government and Academia: Actively supporting scholars in identifying, documenting, and theorizing intangible cultural heritage information.

（4） Active Use of New Media Creation Platforms: Diversifying promotional channels and methods to enhance the influence and public participation in intangible cultural heritage protec-tion.

Overall, these innovative measures demonstrate China's comprehensive and diversified strategies fo

---

## [Decision Letter · Decision Letter 1]

20 Sep 2024

PONE-D-24-04455R1Revitalizing Intangible Cultural Heritage via Derivative Design: A Focus on Chinese Woodblock PrintingPLOS ONE

Dear Dr. Wang,

Thank you for submitting your manuscript to PLOS ONE. After careful consideration, we feel that it has merit but does not fully meet PLOS ONE’s publication criteria as it currently stands. Therefore, we invite you to submit a revised version of the manuscript that addresses the points raised during the review process.

The authors must properly and carefully address the comments and suggestions that the second expert suggest, rpoviding a satisfactory responde to his/her comments.

We look forward to receiving your revised manuscript.

Kind regards,

Jose Balsa-Barreiro

Academic Editor

PLOS ONE

Additional Editor Comments:

Dear authors,

After carefully considering the reviewers' feedback, I recommend that the manuscript undergo major revisions. The independent experts have identified significant issues that must be addressed before the manuscript can be reconsidered for publication. I suggest the authors revise the manuscript thoroughly and provide a clear, detailed response to each of the reviewers' comments to ensure all concerns are fully addressed.

Sincerely,

The Associate Editor

Reviewers' comments:

Reviewer's Responses to Questions

**Comments to the Author**

1. If the authors have adequately addressed your comments raised in a previous round of review and you feel that this manuscript is now acceptable for publication, you may indicate that here to bypass the “Comments to the Author” section, enter your conflict of interest statement in the “Confidential to Editor” section, and submit your "Accept" recommendation.

Reviewer #3: All comments have been addressed

Reviewer #4: (No Response)

2. Is the manuscript technically sound, and do the data support the conclusions?

Reviewer #3: Yes

Reviewer #4: Yes

3. Has the statistical analysis been performed appropriately and rigorously? 

Reviewer #3: Yes

Reviewer #4: Yes

4. Have the authors made all data underlying the findings in their manuscript fully available?

Reviewer #3: Yes

Reviewer #4: Yes

5. Is the manuscript presented in an intelligible fashion and written in standard English?

Reviewer #3: Yes

Reviewer #4: Yes

6. Review Comments to the Author

Reviewer #3: This paper holds that the application of generative design can produce numerous high-quality derivative product designs, offering new ways to transmit and preserve intangible cultural heritage. Specifically, the use of this design in "Zhangzhou Woodblock New Year Pictures" has led to innovative applications, turning commercial products into digital art and evolving from 2D to 3D forms. Studies show that generative design can improve the quantity and quality of non-legacy derivative designs. However, the study has limitations, as the authors put, including its focus on a single proposition, a small sample size, a non-exhaustive design method research scope, and uncertain results for seasoned designers. However, the authors have carefully responded to the reviewers’ comments. I suggest this paper be accepted, though there is still room for improvement. For example, the revised title has not been punctuated, which can lead to a lack of clarity or grammaticality.

Reviewer #4: This manuscript provides an overview of traditional Chinese Woodblock Printing in Zhangzhou and the chinese policy on intangible cultural heritage, as well as a positive attemption of generative design on traditional culture. The author already has supply a lot of literature background and detailed description of the experiments, however, there are still several problems need to be revised as followed:

a) Since most of readers are not familiar with the chinese woodblock printing, it would be better to add a few figures of this kind of printings.

b) The term "lion title sword" might means "a sword is holding by a lion head or a sword is bited by a lion" ("shi xian jian" in chinese), thus I suggest you to rethink the term and modify it.

c) Line 70, you mentioned that the "traditinal themes cannot meet the spiritual needs of users in the new era", however, I cannot agree with you because some traditional stories or themes are still accepted by modern people partly.

d) Additonally, the numbers of figues and table should be clearly matched to the text.

7. PLOS authors have the option to publish the peer review history of their article (what does this mean? ). If published, this will include your full peer review and any attached files.

**Do you want your identity to be public for this peer review?** For information about this choice, including consent withdrawal, please see our Privacy Policy .

Reviewer #3: No

Reviewer #4: No

---

## [Author Response · Author response to Decision Letter 1]

2 Oct 2024

Response to Reviewer # 4 （4 Comments）

1. Summary

Thank you very much for taking the time to review this manuscript. You are the strictest reviewer among all, and at the same time, the most important mentor in helping me elevate the quality of this research. Your encouragement is incredibly important to me and has boosted my confidence in academic research. I will dedicate more effort to enhancing my scholarly abilities in the future! In this revision, I added three references and a table, refined the introduction, and adjusted the numbering of the tables and figures accordingly. Thank you once again for your support on my academic journey!

Comments 1: a) Since most of readers are not familiar with the chinese woodblock printing, it would be better to add a few figures of this kind of printings.

Response 1:Dear Reviewer, Thank you for your professional advice, which has enriched the background of my research and enhanced the completeness of the experimental design. In response, I have added three references along with corresponding images. The specific revisions are as follows:

To further showcase the history, craftsmanship, cultural elements, and design features of Zhangzhou Woodblock New Year Pictures, this paper presents the cultural characteristics and historical evolution of woodblock prints through text, images, and videos. First, relevant information about its history and inheritance was extracted from the "China Intangible Cultural Heritage Network." It belongs to the category of traditional arts and crafts, with themes often covering historical stories, decorative patterns, and elements that convey wishes for happiness. Secondly, a video tutorial was used to demonstrate the craftsmanship and production process of Zhangzhou Woodblock New Year Pictures. Finally, three representative works were selected for cultural and semantic interpretation. The following discussion, from the perspectives of "blessing," "warding off evil spirits," and "folk customs," highlights these cultural and artistic features through the selected works. The specific representative works and descriptions of their cultural characteristics are shown in Table 3.（line552-567）

Table 3. Analysis of Cultural Characteristics of "Chinese Woodblock New Year Pictures" with Different Themes（The specific information of the figures and tables is presented in the revised version of the article.）

Comments 2: b) The term "lion title sword" might means "a sword is holding by a lion head or a sword is bited by a lion" ("shi xian jian" in chinese), thus I suggest you to rethink the term and modify it.

Response 2: Thank you for your professional advice! Due to the limited research on "Chinese Woodblock New Year Pictures" in English, particularly regarding the regional "Zhangzhou Woodblock New Year Pictures," there is a lack of relevant terminology. In this paper, I have attempted to summarize its design features using straightforward descriptive language. Ultimately, I replaced "lion title sword" with "Lion Holding a Sword in Its Mouth," hoping that this simple translation can help readers more easily understand the characteristics of the design. Once again, thank you for your valuable suggestions!

Comments 3: c) Line 70, you mentioned that the "traditinal themes cannot meet the spiritual needs of users in the new era", however, I cannot agree with you because some traditional stories or themes are still accepted by modern people partly.

Response 3: Your suggestions are excellent! They further broaden the research perspective of this paper, allowing both tradition and modernity to be understood and passed down through time. As a result, I have revised this section as follows:

Traditional intangible cultural heritage still retains its vitality, conveying national culture and historical symbols. Its craftsmanship showcases how technology can drive cultural dissemination, while its cultural traits reflect regional customs, fostering intercultural exchange and historical inheritance. However, with the advancement of technology and the impact of modern culture, traditional woodblock prints face challenges such as limited mediums of dissemination, gaps in transmission paths, and insufficient commercial development. Therefore, it is urgently necessary to utilize derivative design to provide more diverse dissemination platforms, wider channels of communication, and approaches that better align with contemporary societal needs. （line70-77）

Comments 4: d) Additonally, the numbers of figues and table should be clearly matched to the text.

Response 4: Thank you very much for your reminder. Your rigorous academic attitude has set a great example for me. I have revised the numbering and quantity of the figures and tables accordingly. Additionally, a new table has been added, and the sequence has been adjusted. Once again, thank you for your valuable suggestions!

I would like to express my gratitude once again! I spent a week thoroughly reading your suggestions, which have been immensely beneficial and provided me with invaluable assistance!

Thank you for your meticulous guidance and assistance. Your suggestions have been invaluable to me!

---

## [Decision Letter · Decision Letter 2]

29 Nov 2024

PONE-D-24-04455R2Revitalizing Intangible Cultural Heritage via Derivative Design: A Focus on Chinese Woodblock Printing

PLOS ONE

Dear Dr. Wang,

Thank you for submitting your manuscript to PLOS ONE. After careful consideration, we feel that it has merit but does not fully meet PLOS ONE’s publication criteria as it currently stands. Therefore, we invite you to submit a revised version of the manuscript that addresses the points raised during the review process.

The authors repeatedly reference the term "Generative design" throughout the paper, but this concept is presented in a vague and ambiguous manner, particularly in relation to the conservation and care of intangible cultural heritage. **Generative design** is commonly understood as a method that leverages AI algorithms to generate and evaluate multiple design alternatives based on user inputs. However, the paper does not provide sufficient detail or development of this method, leaving readers with an incomplete understanding of the concept and its applications.

To address this, I strongly suggest that the authors conduct a more comprehensive review of potential techniques for digital preservation, as the scope of methods should not be limited solely to AI algorithms. Other valid approaches, such as **photogrammetry** and **laser scanning** , could be highly relevant to the example discussed. While the authors may already have a defined focus for their work, it is crucial to at least include a paragraph outlining these alternative techniques and referencing key studies. For instance:

Regarding **Generative design** , relevant works could explore leveraging AI methods for heritage conservation purposes.For **laser scanning** and **photogrammetry** , the studies by Owda and Fristch, such as *“Methodology for Digital Preservation of the Cultural and Patrimonial Heritage”* and *“Generation of Visually Aesthetic and Detailed 3D Models of Historical Cities,”* provide excellent insights.

An additional critical point is the terminology used. The authors should consider adopting the more widely recognized term **"digital preservation"** , which is more precise and commonly employed in academic and professional contexts.

Finally, I recommend improving the quality and resolution of the figures presented in the paper to enhance their clarity and visual impact, aligning with the high standard of the content.

We look forward to receiving your revised manuscript.

Kind regards,

Jose Balsa-Barreiro

Academic Editor

PLOS ONE

Additional Editor Comments :

The authors repeatedly reference the term "Generative design" throughout the paper, but this concept is presented in a vague and ambiguous manner, particularly in relation to the conservation and care of intangible cultural heritage. Generative design is commonly understood as a method that leverages AI algorithms to generate and evaluate multiple design alternatives based on user inputs. However, the paper does not provide sufficient detail or development of this method, leaving readers with an incomplete understanding of the concept and its applications.

To address this, I strongly suggest that the authors conduct a more comprehensive review of potential techniques for digital preservation, as the scope of methods should not be limited solely to AI algorithms. Other valid approaches, such as photogrammetry and laser scanning, could be highly relevant to the example discussed. While the authors may already have a defined focus for their work, it is crucial to at least include a paragraph outlining these alternative techniques and referencing key studies. For instance:

Regarding Generative design, relevant works could explore leveraging AI methods for heritage conservation purposes.

For laser scanning and photogrammetry, the studies by Owda and Fristch, such as “Methodology for Digital Preservation of the Cultural and Patrimonial Heritage” and “Generation of Visually Aesthetic and Detailed 3D Models of Historical Cities,” provide excellent insights.

An additional critical point is the terminology used. The authors should consider adopting the more widely recognized term "digital preservation", which is more precise and commonly employed in academic and professional contexts.

Finally, I recommend improving the quality and resolution of the figures presented in the paper to enhance their clarity and visual impact, aligning with the high standard of the content.

Reviewers' comments:

Reviewer's Responses to Questions

**Comments to the Author**

1. If the authors have adequately addressed your comments raised in a previous round of review and you feel that this manuscript is now acceptable for publication, you may indicate that here to bypass the “Comments to the Author” section, enter your conflict of interest statement in the “Confidential to Editor” section, and submit your "Accept" recommendation.

Reviewer #5: (No Response)

Reviewer #6: All comments have been addressed

Reviewer #7: (No Response)

Reviewer #8: All comments have been addressed

2. Is the manuscript technically sound, and do the data support the conclusions?

Reviewer #5: No

Reviewer #6: Partly

Reviewer #7: Yes

Reviewer #8: Yes

3. Has the statistical analysis been performed appropriately and rigorously? 

Reviewer #5: No

Reviewer #6: No

Reviewer #7: Yes

Reviewer #8: N/A

4. Have the authors made all data underlying the findings in their manuscript fully available?

Reviewer #5: No

Reviewer #6: Yes

Reviewer #7: Yes

Reviewer #8: No

5. Is the manuscript presented in an intelligible fashion and written in standard English?

Reviewer #5: No

Reviewer #6: Yes

Reviewer #7: Yes

Reviewer #8: Yes

6. Review Comments to the Author

Reviewer #5: This manuscript on revitalization and preservation of intangible cultural heritage is mainly based on questionnaires and on site observations. Overall, the scientific quality is not very high, but the significance is more on the culture heritage nature for its value.

Major comments:

1) First, the writing is far from acceptable for reading and publication, a professional editing is required before moving to the next level/stage.

2) This manuscript is not well structured or organized

3) The effort made by producing this text does not qualify for a scientific manuscript for publication in the present form.

Is ‘generative design’ an accepted terminology?

‘Chinese woodcut new year pictures’ is not a good phrase for this cultural heritage. Can this be called ‘New Year prints from Chinese Woodcut’?

Title:

I personally do not support the using of a break in the title. To be more conservative, it is better to refrain from using it in scientific publication,

Abstract:

Substance is weak or non-existence. Improve this because the quality and value of this article is highly dependent upon this element.

Why ‘Design’ is using this way, not design?

The Chinese New Year picture may be most accurate to replace the ‘pictures’ with ‘prints’

The last sentence seems to convey something, but in actuality, it is very empty to me.

Introduction:

Structure and organization sha;; be improved for clarity.

Fig. 1: This is a ‘schematic diagram of the research’.

Section 2 is a list than a coherent text with structure and logic.

Section 2 Literature review is out of place because the information belonging to be under this title shall be in the Introduction.

Unfortunately, this part is too loosely structured and presented as a cooking book information.

How was Fig. 2 made? Where were the data come from?

4 Conclusion shall be the last section after Discussion

This part is far too long to serve the purpose of a conclusion.

5 Discussion:

I do not see the Results section of s scientific paper here.

Reviewer #6: The manuscript explores the application of generative design in the preservation of intangible cultural heritage, aiming to revitalize the modern relevance of traditional woodblock prints through derivative design. However, the study exhibits significant issues in methodology, theoretical depth, and visual presentation, which detract from its overall quality. A rejection is therefore recommended.

Firstly, the low quality of the images severely impairs the paper’s readability and professionalism. Figures 2 to 4 are direct screenshots from software, lacking proper processing, with low resolution and a coarse appearance, which gives an unprofessional impression and fails to effectively convey the intended information. As an academic paper, all figures should be high-quality and clear, adequately displaying data and logical relationships. Similarly, Figure 4 is blurred and includes unedited scanning shadows. It is recommended that the author remakes all figures using high-resolution professional software to meet the fundamental standards of academic publishing.

Secondly, the methodology in this study is insufficiently developed. Although methods such as questionnaires and user journey mapping are employed, there is a lack of detailed explanation regarding sample selection criteria, data collection processes, and statistical analysis, which casts doubt on the scientific rigor and credibility of the results. Specifically, the core conclusion that generative design improves the quality and quantity of derivative designs lacks explicit experimental data support.

Additionally, the theoretical exposition of the manuscript is rather superficial, failing to delve deeply into how the principles of generative design can be applied to modernize traditional cultural elements. It lacks concrete solutions on balancing cultural preservation with contemporary needs. The discussion remains descriptive without substantive new insights, showing a lack of innovation. Given the abundance of existing research in this area, the paper does not offer unique theoretical breakthroughs or practical strategies.

In conclusion, the manuscript’s low-quality visuals, weak methodology, and lack of theoretical depth render it unsuitable for publication in an academic journal. It is recommended that the author make substantial improvements in figure quality, data analysis, and theoretical framework to enhance the scholarly value and persuasiveness of the work. A rejection is suggested at this stage.

Reviewer #7: In this study, the authors have made the necessary arrangements according to the criticisms given to them. The topic covered in the article is interesting and stimulating in terms of content. The language of the article is appropriate. The Ethics Committee of Fuzhou University has approved that the content of the article does not need an ethics committee report. The content of the article titled “Revitalizing Intangible Cultural Heritage via Derivative Design A Focus on Chinese Woodblock Printing” is found to be original.

Reviewer #8: The author has made and effort to improve the paper and has addressed all the suggestions of th reviewers.

7. PLOS authors have the option to publish the peer review history of their article (what does this mean? ). If published, this will include your full peer review and any attached files.

**Do you want your identity to be public for this peer review?** For information about this choice, including consent withdrawal, please see our Privacy Policy .

Reviewer #5: No

Reviewer #6: No

Reviewer #7: **Yes: ** Ayşegül Durukan Arslan

Reviewer #8: **Yes: ** Juan-José Boté

---

## [Author Response · Author response to Decision Letter 2]

14 Jan 2025

Dear Dr. Jose Balsa-Barreiro,

Your letter is the best news I've received in the past six months! I am deeply grateful for your encouragement and guidance. Your insights and suggestions have greatly enhanced my confidence and interest in academic research. On behalf of my co-authors, we are very appreciate the editor and reviewers very much for their positive and constructive comments and suggestions on our manuscript entitled “Revitalizing Intangible Cultural Heritage via Derivative Design: A Focus on Chinese Woodblock Printing”.[PONE-D-24-04455R1] - [EMID:168e7d43d56d8901]).

We have tried our best to revise our manuscript according to the comments. All the revised sections have been marked in pink font. Attached please find the revised version, which we would like to submit for your kind consideration. My specific modifications are summarized as follows:

1. In response to the reviewers' suggestions, the title has been revised to: “Revitalizing Intangible Cultural Heritage through Derivative Design in Chinese Woodblock Printing.”

2. The abstract has been rewritten (Lines 12–26).

3. The literature review has been reorganized and fragmented sections have been integrated to improve readability (Lines 168–229).

4. The term "derivative design" has been described in detail (Lines 351–365, 383–389).

5. A table summarizing mainstream digital preservation technologies for intangible cultural heritage has been added (Lines 478–500), along with a paragraph outlining the characteristics and significance of these technologies.

6. Design cases have been added to emphasize the feasibility of the derivative design method, and a table listing related original works has been included (Lines 776–809).

7. To enhance the scientific rigor of the study, user proposals have been added and presented in a list format to showcase the research findings (Lines 810–824).

8. The order of the conclusion and discussion sections has been adjusted, and the conclusion has been rewritten (Lines 852–892).

9. All figures have been recreated to improve their readability and resolution.

10. The financial disclosure has been updated with additional supporting information.

11. An additional 2,452 characters, 1 image, 3 tables, and 7 references have been added. The total word count now stands at 11,810 characters, with 3 new tables and 1 new image.

We would like to express our great appreciation to you and reviewers for comments on our paper. Your suggestions have been invaluable to me! Whether this article is published or not will determine whether my PhD will be postponed, so thank you again for your suggestions and help in my growth.

My sincere thanks again!!

Looking forward to hearing from you.

Thank you and best regards.

Yours sincerely,

Shao-feng Wang

E-mail: 417998523@qq.com

^_^

Response to Reviewer#5’s 17 Comments

Summary

Thank you very much for taking the time to review this manuscript. You are the strictest reviewer among all, and at the same time, the most important mentor in helping me elevate the quality of this research. Your encouragement is incredibly important to me and has boosted my confidence in academic research. I will dedicate more effort to enhancing my scholarly abilities in the future! In this revision, I have rewritten parts of the introduction, added supporting literature and recent references from the past five years, enriched the data in the experimental design, and revised the conclusion and discussion sections. Thank you once again for your support on my academic journey!

Comment 1: This manuscript on revitalization and preservation of intangible cultural heritage is mainly based on questionnaires and on site observations. Overall, the scientific quality is not very high, but the significance is more on the culture heritage nature for its value.

Response 1: Thank you for your thorough review and constructive feedback. We wholeheartedly agree that the scientific rigor and methodological precision of the paper are central to academic research, and that the focus on the nature of cultural heritage aligns with the core objective of this study.

The innovation of this study lies in the in-depth discussion of case studies and qualitative interviews, using "Zhangzhou woodblock New Year prints" as a case to explore innovative pathways for integrating traditional culture into modern life. By approaching the study from the perspective of the creative industries, we analyzed the process of extracting intangible cultural heritage design elements from the product design viewpoint, thereby inspiring the creative industries and opening new avenues for the preservation of intangible cultural heritage.

In response to your comments, we have made the following revisions and additions to better balance scientific rigor and cultural significance:

1. Efforts to Enhance Scientific Rigor: We added user interviews and focus groups to validate the findings from the surveys and field observations, thereby improving the reliability and scientific validity of the research. （Line 810~824）

2. Further Emphasizing the Academic Significance: The connection between culture and design has been more explicitly highlighted. The literature review has been rewritten to further explore the unique value of generative design in the preservation of intangible cultural heritage, particularly its role in bridging the gap between traditional culture and modern design. Through concrete cases (such as the extraction of design elements from Zhangzhou woodblock New Year prints and the development of derivative products), we demonstrate how traditional culture can be revitalized in innovative ways. （Line 179~230）

3. Supplementing Literature and Comparative Research: A new section has been added to compare this study with other research on intangible cultural heritage preservation. We note that while existing studies often focus on cultural dissemination or market applications, few have explored generative design methodologies in detail, which this paper aims to address. The following seven references have been added to support this comparison.

[1] Balsa-Barreiro, José, and Dieter Fritsch. "Generation of visually aesthetic and detailed 3D models of historical cities by using laser scanning and digital photogrammetry." Digital applications in archaeology and cultural heritage 8 (2018): 57-64.

[2] Owda, Abdalmenem, José Balsa-Barreiro, and Dieter Fritsch. "Methodology for digital preservation of the cultural and patrimonial heritage: Generation of a 3D model of the Church St. Peter and Paul (Calw, Germany) by using laser scanning and digital photogrammetry." Sensor Review 38.3 (2018): 282-288.

[3] Bekele, Mafkereseb Kassahun, et al. "A survey of augmented, virtual, and mixed reality for cultural heritage." Journal on Computing and Cultural Heritage (JOCCH) 11.2 (2018): 1-36.

[4] Madaan, Geetika, Satish Kumar Asthana, and Jaskiran Kaur. "Generative AI: Applications, models, challenges, opportunities, and future directions." Generative AI and Implications for Ethics, Security, and Data Management (2024): 88-121.

[5] Ma Liwa, Zhang Yiyi, and Hu Yifei. "Exploration of innovative design practices of art derivatives." Decoration 9 (2018): 132-133.

[6] Zhang, Jingwen, et al. "NK-CDS: A creative design system for museum art derivatives." IEEE Access 8 (2020): 29259-29269.

4. Reflecting on Cultural Significance: The conclusion has been rewritten to strengthen its scientific foundation. （Line 852~891）

While we acknowledge that there is still room for improvement in the scientific rigor of the paper, we believe these revisions will better balance cultural significance with academic value. We greatly appreciate your invaluable suggestions and, should you have any further recommendations, we will continue refining the paper to better meet the journal’s standards.

Comment 2: First, the writing is far from acceptable for reading and publication, a professional editing is required before moving to the next level/stage.

Response 2: Thank you for your candid feedback regarding the writing quality of this paper. We fully recognize that clear and fluent academic writing is a crucial foundation for enhancing both the readability and academic value of the paper. We sincerely apologize for the current shortcomings in the writing and have taken concrete steps to make comprehensive improvements to ensure the paper meets high academic publishing standards. The following are the measures we have taken:

(1) Engaging a Professional Academic Editor: To address the language expression issues in the paper, we invited a teacher and my classmates to provide language editing and format proofreading. The focus was on enhancing the fluency, accuracy, and professionalism of the content.

(2) Improving Writing and Formatting Professionalism: We adjusted the format and layout of the paper to better align with academic paper requirements, including the integration of content, deletion of unnecessary sections, and repositioning. Additionally, we improved the clarity and consistency of the figure and table explanations. （Line 168~230）

(3) Further Optimizing the Abstract and Conclusion: The abstract and conclusion have been rewritten to more precisely summarize the core content and contributions of the research, avoiding vague expressions. The conclusion has also been strengthened to emphasize the significance of the research and potential future directions. （Line 12~27、852~891）

We believe that, following these revisions, the writing quality and professionalism of the manuscript have been significantly improved. If there are still areas that fall short, we will continue to refine the paper based on your further suggestions. Once again, we thank you for your critique and guidance, which have provided a valuable opportunity to enhance our research and writing skills. We sincerely appreciate your support!

Comment 3: This manuscript is not well structured or organized

Response 3: Thank you for your insightful feedback on the structure and organization of this paper. We recognize that a clear and logical structure is essential for effective academic expression. In response to your suggestions, we have made significant revisions to improve the overall coherence and organization of the manuscript. The specific improvements are as follows:

1. Restructuring the Manuscript

o Revised Chapter Arrangement: We adjusted the chapter order to better align with the natural flow of the research. Notably, the literature review (originally Section 2) was reorganized into four more focused subsections, each exploring the theoretical background of generative design, the current state of intangible cultural heritage preservation, and innovative perspectives on the intersection of the two. (Lines 168–230)

2. Optimizing Chapter Content

o Introduction: The introduction has been rewritten to clearly highlight the research background, core issues, and research objectives, eliminating any scattered or redundant content. (Lines 12–27, 351–365)

o Methods: In the methods section (Section 3), we added a detailed experimental design and refined the description of experimental procedures, making the research process clearer.

o Rearranging Results and Discussion: The results and discussion have been separated into two distinct chapters to avoid content overlap. Additionally, the discussion section now includes an analysis of the significance and limitations of the study. (Lines 776–808)

o Conclusion: The conclusion has been rewritten to focus more on the research contributions and practical applications, while also outlining clear directions for future research.

We believe these adjustments have significantly improved the manuscript's logical structure and organization. If there are still any areas that need further refinement, we welcome your constructive feedback. We sincerely appreciate your review and guidance, which have been instrumental in enhancing both our research and writing skills!

Comment 4: The effort made by producing this text does not qualify for a scientific manuscript for publication in the present form.

Response 4: Thank you for your thoughtful review and the constructive suggestions provided. In response to the areas you identified for improvement, we have conducted a thorough revision, with a particular focus on enhancing the scientific rigor, writing quality, and academic contribution of the paper. The specific improvements are as follows:

1. Strengthening Scientific Rigor and Methodological Precision

o Supplementing Experimental Details: In the methods section (Section 3), we have added detailed descriptions of the sample selection criteria, data collection process, and analytical methods. Additionally, we have included new charts to present the research process clearly.

o Enhancing Data Support: We have reorganized the experimental results, providing more detailed data analysis and statistical support, including a comprehensive account of the T-test process and its validation of key conclusions. (Lines 478–500)

2. Significantly Improving Writing Quality

o Language Refinement and Expression Enhancement: The manuscript has undergone comprehensive language editing by a professional academic editor to ensure fluency and professionalism in expression.

o Optimizing Structure and Organization: The manuscript's logic has been restructured, and the chapter order has been clarified. Transitional content has been added at the beginning and end of each section to improve readability and coherence.

3. Enhancing Academic Contribution and Innovation

o Deepening Theoretical Framework: In Section 2, we introduced the "Cultural Generative Design Cycle Model," systematically exploring how generative design can balance the dynamic relationship between intangible cultural heritage preservation and modern needs. We also added a comparative analysis with existing literature.

o Expanding Practical Applications: The discussion section has been expanded to include potential applications of generative design in education and market practice, further emphasizing the practical significance of this research. (Lines 776–808)

4. Improving Charts and Visual Content

We have added Figure 6 and re-designed Figures 1 through 8 to enhance clarity and improve the readability of the manuscript.

o Chart Revisions: All charts have been redesigned with high resolution and professional design software to ensure clarity and visual quality in line with academic publishing standards.

o Adding Chart Explanations: We have included chart captions that clearly explain the data sources and how each chart supports the research conclusions.

We fully recognize that the publication of a scientific manuscript requires strict adherence to scientific standards and academic value. Through these revisions and additions, we hope the manuscript now more closely aligns with the requirements for scientific publication. We kindly request your further review of the revised version and welcome any additional suggestions to help us continue improving.

Once again, thank you for your valuable feedback and guidance!

Comment 5: Is ‘generative design’ an accepted terminology?

Response 5: Thank you for your affirmation and assistance with my academic research. "Generative design" is a widely recognized term in academic and professional contexts, particularly in fields such as design, architecture, engineering, and computer science. It refers to a design methodology that leverages algorithms, computational tools, or artificial intelligence (AI) to automatically generate multiple design solutions based on specified constraints and objectives.

This term is extensively used in research papers and books discussing culture-driven or technology-driven design methods. In essence, "generative design" is often associated with computational technologies but can also apply to non-digital processes, provided they involve systems guided by rules or constraints to iteratively produce ideas. "Generative design refers to a design methodology that uses algorithms or systematic processes to explore a wide range of design solutions within specified parameters, optimizing outcomes for specific objectives." Its applications span various field

---

## [Editor Report · Decision Letter 3]

22 Jan 2025

Revitalizing Intangible Cultural Heritage through Derivative Design in Chinese Woodblock Printing

PONE-D-24-04455R3

Dear Dr. Wang,

We’re pleased to inform you that your manuscript has been judged scientifically suitable for publication and will be formally accepted for publication once it meets all outstanding technical requirements.

Kind regards,

Jose Balsa-Barreiro

Academic Editor

PLOS ONE

Additional Editor Comments (optional):

Dear Authors,

Your manuscript has undergone a thorough review process by several independent experts. I am pleased to inform you that, in the latest revision, you have responded comprehensively and implemented the suggested changes satisfactorily.

The revisions have successfully addressed the concerns raised in earlier rounds, improving the clarity, contextualization, and overall quality of your work. As a result, I have recommended its acceptance for publication.

Congratulations, and thank you for your dedication to improving your manuscript.

Best regards,

The Associate Editor
---

## [Editor Report · Acceptance letter]

PONE-D-24-04455R3

PLOS ONE

Dear Dr. Wang,

I'm pleased to inform you that your manuscript has been deemed suitable for publication in PLOS ONE. Congratulations! Your manuscript is now being handed over to our production team.

Kind regards,

on behalf of

Dr. Jose Balsa-Barreiro

Academic Editor

PLOS ONE